# Structural basis for GSDMB pore formation and its targeting by IpaH7.8

Chengliang Wang[1], Sonia Shivcharan[1,4], Tian Tian[1,4], Skylar Wright[1,4], Danyang Ma[1], JengYih Chang[2], Kunpeng Li[3], Kangkang Song[2], Chen Xu[2], Vijay A. Rathinam[1] & Jianbin Ruan[1✉]

Gasdermins (GSDMs) are pore-forming proteins that play critical roles in host defence through pyroptosis[1,2]. Among GSDMs, GSDMB is unique owing to its distinct lipid-binding profile and a lack of consensus on its pyroptotic potential[3–7]. Recently, GSDMB was shown to exhibit direct bactericidal activity through its pore-forming activity[4]. *Shigella*, an intracellular, human-adapted enteropathogen, evades this GSDMB-mediated host defence by secreting IpaH7.8, a virulence effector that triggers ubiquitination-dependent proteasomal degradation of GSDMB[4]. Here, we report the cryogenic electron microscopy structures of human GSDMB in complex with *Shigella* IpaH7.8 and the GSDMB pore. The structure of the GSDMB–IpaH7.8 complex identifies a motif of three negatively charged residues in GSDMB as the structural determinant recognized by IpaH7.8. Human, but not mouse, GSDMD contains this conserved motif, explaining the species specificity of IpaH7.8. The GSDMB pore structure shows the alternative splicing-regulated interdomain linker in GSDMB as a regulator of GSDMB pore formation. GSDMB isoforms with a canonical interdomain linker exhibit normal pyroptotic activity whereas other isoforms exhibit attenuated or no pyroptotic activity. Overall, this work sheds light on the molecular mechanisms of *Shigella* IpaH7.8 recognition and targeting of GSDMs and shows a structural determinant in GSDMB critical for its pyroptotic activity.

Gasdermin (GSDM) proteins consist of an N-terminal pore-forming domain (GSDM-N), a C-terminal autoinhibitory domain (GSDM-C) and an interdomain linker[7–15]. Human GSDMB has multiple splicing isoforms (isoforms 1–6, Q8TAX9; UniProt) varying in their interdomain linkers. Isoforms 1, 4 and 6 contain 'canonical' interdomain linkers whereas isoforms 2 and 3 contain truncated linkers and isoform 5 comprises only the C-terminal domain[16]. While whether GSDMB induces pyroptosis remains controversial, a recent study showed that GSDMB preferentially targets the bacterial membrane and restricts microbial growth and spread directly during infection[4,13]. This process, however, is subverted by *Shigella*—a highly infectious Gram-negative bacterium that causes acute gastroenteritis[17]—through its secreted effector protein, IpaH7.8 (ref. [4]). Surprisingly, IpaH7.8 also binds GSDMD in addition to GSDMB. However, IpaH7.8 ubiquitinates only human, but not mouse, GSDMD[18], which may be related to the fact that humans and non-human primates are the natural reservoirs for *Shigella* whereas mice are not[19]. The molecular mechanisms of why GSDMB functions distinctly from other GSDMs in inducing pyroptosis, and how GSDMB and human GSDMD are specifically recognized by *Shigella* IpaH7.8, are unknown.

### Structure of the GSDMB–IpaH7.8 complex

To understand how GSDMB is recognized by *Shigella* IpaH7.8, we determined the cryogenic electron microscopy (cryo-EM) structure of human GSDMB (isoform 1, Q8TAX9-1) in complex with *Shigella flexneri* IpaH7.8 at an overall resolution of 3.8 Å (Extended Data Fig. 1 and Extended Data Table 1). The structure shows a 1:1 complex with the IpaH7.8-LRR domain binding to GSDMB-N (Fig. 1a,b). IpaH7.8-LRR organizes into a slightly curved solenoid structure containing nine LRR motifs capped by two N-terminal α-helices (α1 and α2) and a C-terminal α4 helix, together with a parallel β10-strand that directly augments the β-sheet of LRR (Fig. 1c). The overall architecture of IpaH7.8-LRR is highly similar to the previously reported structures of LRR domains in IpaH1.4 and IpaH3 (refs. [20,21]) (Fig. 1d). The density of the IpaH7.8 C-terminal NEL domain is not visible on the map, probably due to its flexibility in the complex, allowing for ubiquitination of multiple lysines in GSDMB[4] (Extended Data Fig. 1c).

GSDMB adopts an auto-inhibited conformation very similar to that of GSDMA3 and GSDMD[7,8] (Fig. 1e). The predicted β4 strand (in reference to the GSDMA3 structure) was not observed in the density, whereas the α4 helix protrudes away from the GSDMB-N to contact GSDMB-C (Fig. 1e). The local resolution of GSDMB-C is relatively lower than the GSDMB-N/IpaH7.8-LRR core, indicating dynamics between the N- and C-terminal domains of GSDMB (Extended Data Fig. 1c). Despite its low resolution, we were able to trace the main chain of GSDMB-C, which adopts a bundle consisting of eight helices similar to previously determined crystal structures[6] (Fig. 1e and Extended Data Fig. 2a). GSDMB was thought to have reduced auto-inhibition in its full-length structure

[1]Department of Immunology, School of Medicine, University of Connecticut Health Center, Farmington, CT, USA. [2]Department of Biochemistry & Molecular Biotechnology and Cryo-Electron Microscopy Core Facility, University of Massachusetts Chan Medical School, Worcester, MA, USA. [3]Cryo-Electron Microscopy Core, Case Western Reserve University School of Medicine, Cleveland, OH, USA. [4]These authors contributed equally: Sonia Shivcharan, Tian Tian, Skylar Wright. ✉e-mail: ruan@uchc.edu

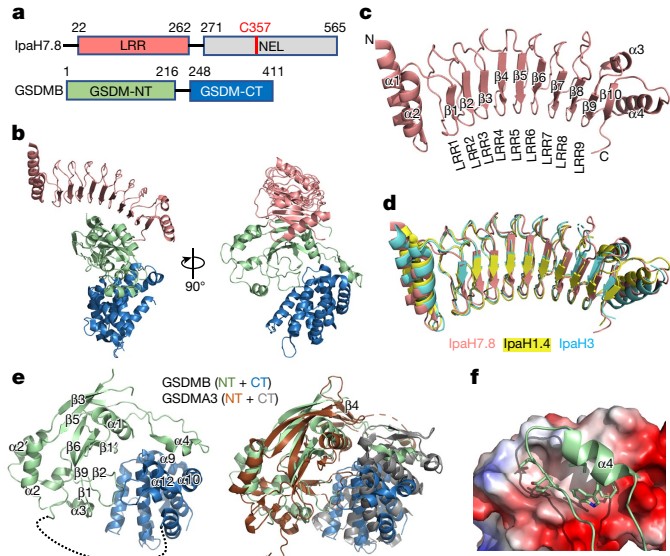

**Fig. 1 | Cryo-EM structure of the GSDMB–IpaH7.8 complex. a**, Domain schemes of IpaH7.8 and GSDMB. The domain boundaries are labelled. **b**, Overall structure of the GSDMB–IpaH7.8 complex in two views. The LRR domain in IpaH7.8 and the GSDMB-N- and -C-terminal domains are coloured as in **a**. **c**, Ribbon diagram showing the structure of the IpaH7.8-LRR domain. **d**, Superposition of IpaH7.8-LRR (salmon) onto IpaH1.4 (yellow; PDB: 7V8H) and IpaH3 (cyan; PDB: 3CVR) LRR domains. **e**, Overall structure of GSDMB (left), and superposition of the structures of GSDMB and mouse GSDMA3 (PDB: 5B5R) (right). The β4 strand in mouse GSDMA3 is not observed in the GSDMB structure. GSDMA3 N- and C-terminal domains coloured brown and grey, respectively. **f**, Close-up view of the interface between GSDMB-N-terminal α4 helix and GSDMB-C. GSDMB-C is shown as electrostatic potentials and the GSDMB-N-terminal α4 helix as a cartoon, with hydrophobic residues involved in the interaction shown as sticks. NT, N-terminal domain; CT, C-terminal domain.

because of (1) its ability to bind phosphatidylinositol phosphates and sulfatides without cleavage and (2) its lack of a subdomain in GSDMB-C that is essential for interdomain interactions[6,22] (Extended Data Fig. 2b). In our structure, GSDMB-C traps the protruded GSDMB-N α4 helix through a gigantic hydrophobic groove formed by helices α9, α10 and α12 rather than by helices α9 and α11 contributed by the subdomain in GSDMA3, resulting in an even larger interdomain interface in GSDMB (3,020 Å² total interface area in GSDMB versus 2,745 Å² in GSDMA3) (Fig. 1e,f). We speculate that full-length GSDMB might use a different mechanism for phospholipid binding other than cleavage.

## Interactions between GSDMB and IpaH7.8

In the GSDMB–IpaH7.8 complex, six out of nine LRR motifs in IpaH7.8-LRR interact with GSDMB-N, resulting in a total buried surface area of about 1,826 Å² between the two proteins, with key interaction areas divisible into two patches, I and II. LRR7–LRR9 of IpaH7.8 contact a short loop (α1–β1' loop, residues 15–21) containing several negatively charged residues in GSDMB-N to form the first patch (I), whereas IpaH7.8-LRR4–6 interact with the β3 strand in the first extension domain (ED1)[9,23] in GSDMB forming the second patch (II) (Fig. 2a).

Interface patch I is mainly mediated by polar interactions. In particular, side chains of Q185 and R186 from LRR7, Y207 and H209 from LRR8, and R228, N230 and S232 from LRR9 in IpaH7.8 form ten hydrogen bonds with E15, D17, A18 and D21 in GSDMB. Meanwhile, the positively charged side chains of R186 and R228 in IpaH7.8 form extra salt bridges with the negatively charged side chains of E15 and D21, respectively, in GSDMB (Fig. 2b). Interestingly, the side chains of E15 and D17 in GSDMB insert

into two small basic pockets formed by R186 and H209 together with the surrounding residues in IpaH7.8 (Extended Data Fig. 3a). The two pockets are approximately 10 Å apart, suggesting a potential mechanism in which IpaH7.8 recognizes a specific motif containing two acidic residues at this precise distance. Indeed, single mutations of either E15 or D17 in GSDMB into a reversely charged or uncharged residue significantly or partially reduced ubiquitination of GSDMB by IpaH7.8 in vitro (Fig. 2c).

Interaction patch II is mediated by hydrogen bonds and hydrophobic and Van der Waals interactions. The side chain of R125 from IpaH7.8-LRR4 forms two hydrogen bonds with the backbone carbonyl group of E83 and carboxylate oxygen OE1 of Q85 in GSDMB. In parallel, the side chain of Y165 from LRR6 forms two hydrogen bonds with the backbone amino group of L87 and the backbone carbonyl group of Q85 in GSDMB, and Y166 from LRR6 forms a hydrogen bond with the backbone carbonyl group of L87 in GSDMB. Additionally, residues F143, E145 and N146 in LRR5, and F161 and H163 in LRR6 of IpaH7.8, contact the GSDMB β3 strand through extensive hydrophobic and Van der Waals interactions (Fig. 2d).

Structure-based sequence alignment shows that key residues in both patches in IpaH7.8 are divergent in the equivalent positions in other IpaH family proteins (Extended Data Fig. 3b), making IpaH7.8 the unique member in the IpaH family that specifically targets GSDMB[4,18]. Mutations of key residues in IpaH7.8, such as R186E, H209G, R228D and R186E/R228D in interface patch I, and F143S, F161G/I181G, Y165A/Y166A and Y165E/Y166E in interface patch II, all largely or completely abolish the ability of IpaH7.8 to ubiquitinate GSDMB (Fig. 2e). Mutations of R186E/R228D and Y165E/Y166E also significantly attenuate the IpaH7.8-mediated ubiquitination and degradation of GSDMB in cells (Fig. 2f, Extended Data Fig. 3c and Supplementary Figs. 1 and 2).

## IpaH7.8 does not bind mouse GSDMD

*Shigella* IpaH7.8 was previously shown to bind both human and mouse GSDMD; however, it ubiquitinates only human GSDMD[18]. Using isothermal titration calorimetry (ITC), we characterized the interactions between IpaH7.8 and GSDMB/D. IpaH7.8 showed a 46-fold lower affinity for human GSDMD (with a dissociation constant ($K_d$) of 20.4 ± 1.8 μM) than for GSDMB ($K_d$ = 0.44 ± 0.03 μM) (Extended Data Fig. 4a,b). Surprisingly, no interaction was detected between IpaH7.8 and mouse GSDMD (Extended Data Fig. 4c). In agreement with their affinities, analytical gel filtration chromatography showed that IpaH7.8 efficiently comigrated with GSDMB and partially migrated with human GSDMD whereas it barely comigrated with mouse GSDMD (Extended Data Fig. 4d–f). We therefore conclude that *Shigella* IpaH7.8 binds human but not mouse GSDMD.

## The structural determinant in GSDM

Substitution of a fragment near the N terminus (residues 16–21) by the human sequence successfully rescued IpaH7.8-mediated degradation of mouse GSDMD in cells[18]. In our complex structure, this fragment corresponds to the α1–β1' loop in GSDMB in which a motif of three negatively charged residues (α1–β1' motif) interacts extensively with IpaH7.8 (Fig. 2b). Structure-based sequence alignment showed that this motif is conserved in human but not mouse GSDMD ($E_{15}MD_{17}AGGD_{21}$ in GSDMB, $E_{15}LD_{17}HGGD_{21}$ in human GSDMD and $E_{15}VS_{17}GSR_{20}GD_{22}$ in mouse GSDMD) (Extended Data Fig. 2b). Mouse GSDMD contains the conserved E15, but the second acidic residue is substituted by a serine (mS17). Mouse GSDMD also contains the conserved third acidic residue (mD22) but has a unique arginine insertion at position 20 (mR20) (Extended Data Fig. 2b). Substitution of D17 with a serine may disrupt its interaction with Y207 and N230 in IpaH7.8 whereas the mR20 insertion probably pushes the following aspartic acid (mD22) away from interaction with R228 in IpaH7.8 (Fig. 2b), thus blocking interactions between mouse GSDMD and IpaH7.8. Accordingly, a single mutation with an arginine insertion at position 20 (R20Ins) and a double mutation

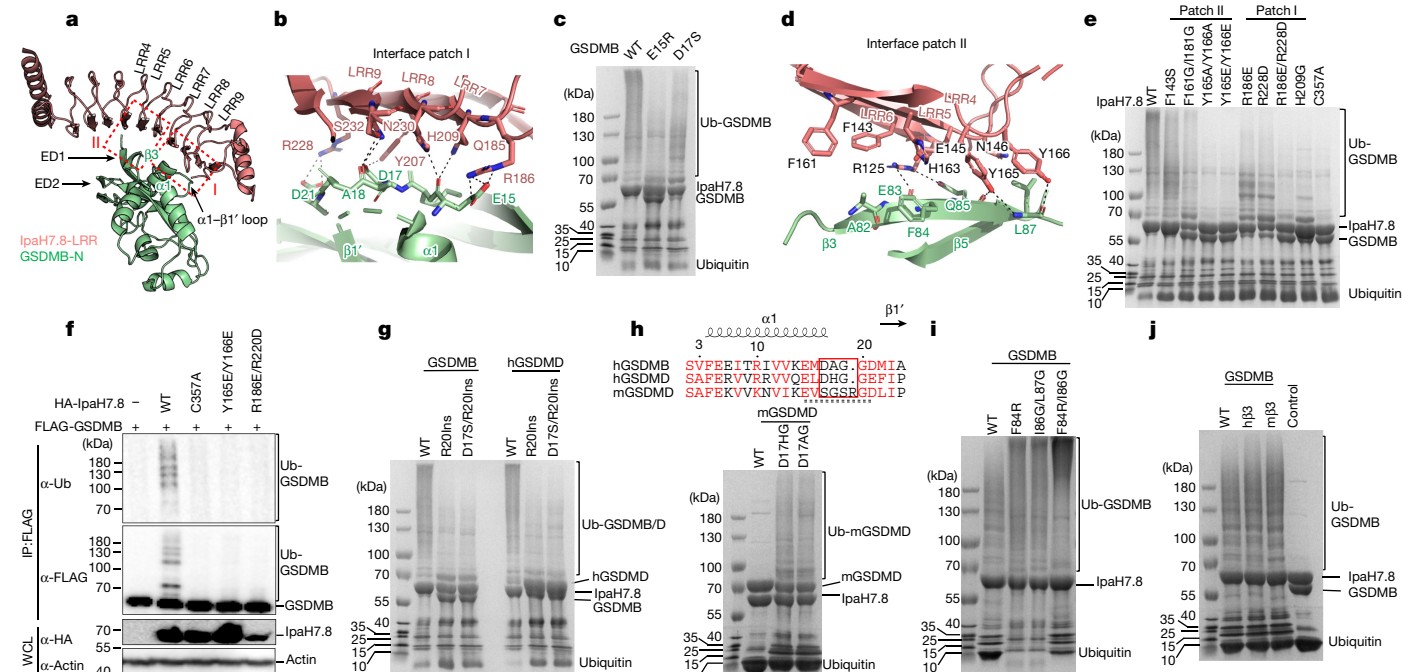

**Fig. 2 | The three-negatively-charged-residue motif is the structural determinant in GSDM recognized by IpaH7.8. a**, The two interfaces between GSDMB and IpaH7.8 are highlighted by dashed boxes. **b**, Detailed interactions in interface patch I. Residues participating in this interface are shown as ball and sticks; hydrogen bonds are shown as grey dotted lines. **c**, Coomassie blue-stained SDS–polyacrylamide gel electrophoresis (SDS–PAGE) of in vitro ubiquitination reactions using GSDMB (WT and mutants) and IpaH7.8. Results representative of three independent experiments. **d**, Detailed interactions in interface patch II. Residues participating in this interface are shown as ball and sticks; hydrogen bonds are shown as grey dotted lines. **e**, Coomassie blue-stained SDS–PAGE of in vitro ubiquitination of GSDMB by IpaH7.8 (WT or mutants). Results representative of three independent experiments. **f**, Immunoblots of GSDMB ubiquitination in HEK293T cells cotransfected with FLAG-tagged GSDMB and HA-IpaH7.8 (WT or mutants). WCL, whole cell lysates. Results representative of three independent experiments. **g**, Coomassie blue-

stained SDS–PAGE of in vitro ubiquitination of GSDMB or human GSDMD (WT and mutants) by IpaH7.8. Results representative of three independent experiments. **h**, Coomassie blue-stained SDS–PAGE of in vitro ubiquitination of mouse GSDMD (WT and mutants) by IpaH7.8. $D_{17}HG$ and $D_{17}AG$ replace the non-conserved sequence ($S_{17}GSR_{20}$) in mouse GSDMD by corresponding human GSDMD and GSDMB sequences, respectively. Amino acid sequence alignment of the α1–β1′ motif (underlined) among GSDMB and human and mouse GSDMD is shown above SDS–PAGE. The swapped sequences among three GSDMs are highlighted by the red box. Results representative of three independent experiments. **i,j**, Coomassie blue-stained SDS–PAGE of in vitro ubiquitination reactions using IpaH7.8 and GSDMB (WT and mutants). The β3 strand in GSDMB was either mutated as indicated in **i** or swapped by the corresponding human (hβ3) and mouse (mβ3) GSDMD sequences in **j**. Control in **j** indicated the ubiquitination of WT GSDMB with mutant IpaH7.8$^{C357}$. Results representative of three independent experiments.

of D17S/R20Ins in GSDMB and human GSDMD blocked their interaction with IpaH7.8 and subsequently compromised IpaH7.8-mediated ubiquitination (Fig. 2g and Extended Data Fig. 4g,h). On the other hand, replacement of the non-conserved α1–β1′ motif in mouse GSDMD with the corresponding GSDMB or human GSDMD residues rescued the ubiquitination of mouse GSDMD by IpaH7.8 (Fig. 2h).

Despite the similar structural feature, the amino acid sequence of β3 strands in GSDMs is highly divergent (Extended Data Fig. 4i). We wondered whether the β3 strand in GSDMs functions as another structural determinant for recognition by IpaH7.8. However, the β3 strand in GSDMB interacts with IpaH7.8 mainly through its backbone, indicating a sequence-independent interaction mode. In agreement with our hypothesis, mutations in the β3 strand of GSMDB did not affect ubiquitination (Fig. 2i). Moreover, swapping β3 strands among GSDMB, human GSDMD, and a mouse GSDMD mutant harbouring a rescue mutation ($D_{17}HG$) did not affect their ubiquitination (Fig. 2j and Extended Data Fig. 4j). We therefore conclude that the three-negatively-charged-residue motif in the α1–β1′ loop is the structural determinant in GSDMs.

## IpaH7.8 binding directly inhibits GSDMB

Because the β3 strand is an essential element of GSDM pore formation[8,9,23], we then tested whether the binding of IpaH7.8 to

this strand directly inhibits GSDMB pore formation. As expected, granzyme A (GZMA)-cleaved GSDMB caused a rapid leakage of 6-carboxyfluorescein (6-FAM) dye from liposomes containing cardiolipin (CL)[4] (Fig. 3a), indicating the formation of GSDMB pore. This GSDMB-induced liposome leakage was largely inhibited in the presence of IpaH7.8, whereas no significant inhibition was observed when GSDMB was incubated with IpaH7.8$^{Y165E/Y166E}$ or IpaH7.8$^{R228D/R186E}$, two IpaH7.8 mutants that no longer interact with GSDMB (Fig. 3a and Extended Data Fig. 5a). IpaH7.8$^{C357A}$, a catalytically inactive mutant that still binds GSDMB[4,18], also attenuated GSDMB-induced liposome leakage (Fig. 3a). Concentration titration further demonstrated that IpaH7.8 inhibited the activity of GSDMB in a dose-dependent manner, with a half-maximal inhibitory concentration ($IC_{50}$) of 1.46 ± 0.11 μM (Fig. 3b and Extended Data Fig. 5b). IpaH7.8 did not affect the cleavage of GSDMB, but directly inhibited the binding of GSDMB-N to liposomes (Extended Data Fig. 5c). Interestingly, IpaH7.8 did not inhibit liposome leakage caused by caspase-11-cleaved human GSDMD (Extended Data Fig. 5d). The association of human GSDMD-N with liposomes was inhibited only by very high concentrations of IpaH7.8 (Extended Data Fig. 5e).

Because GSDMB possesses direct microbiocidal activity by recognition of specific phospholipids on the bacterial membrane[4], we then examined whether IpaH7.8 directly inhibits the antibacterial activity of GSDMB. We found that about 50% of *Escherichia coli* DH5α was killed when exposed to GZMA-cleaved GSDMB, whereas bacterial viability

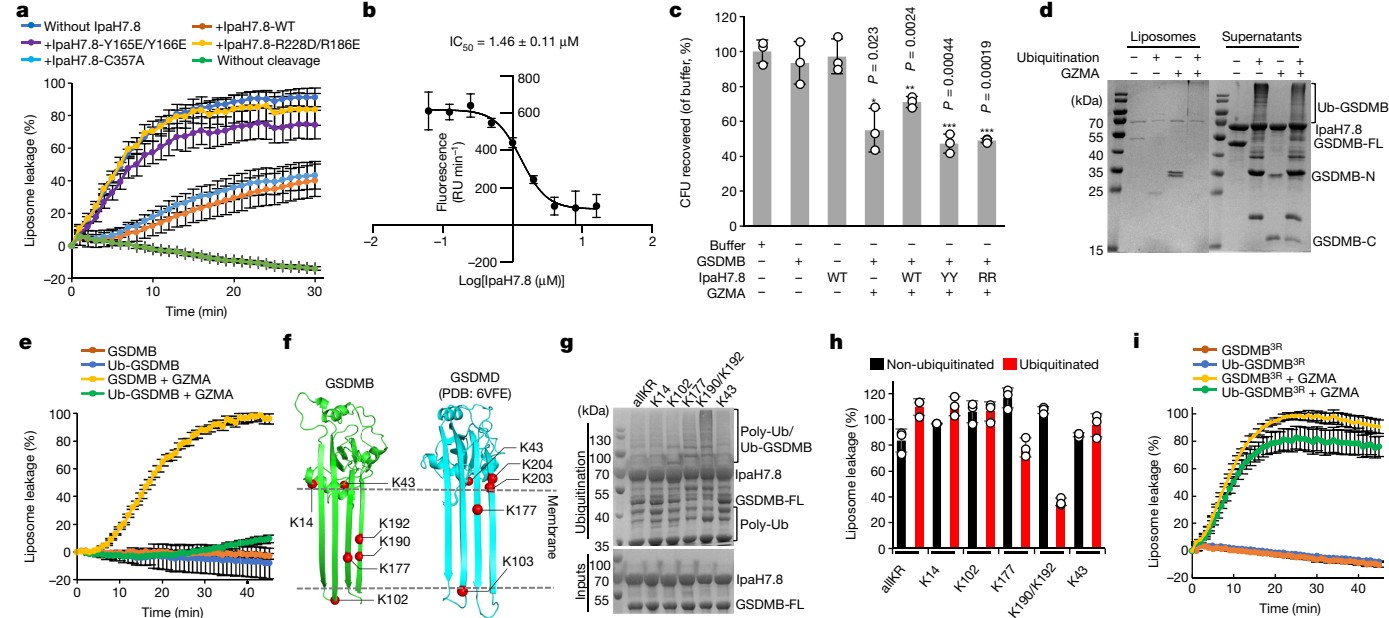

**Fig. 3 | IpaH7.8 inhibits GSDMB pore formation. a**, The ability of GSDMB to induce leakage of CL-containing liposomes (CL-liposomes) when incubated with IpaH7.8 (WT or indicated mutants). **b**, Dose–response curve of IpaH7.8 in inhibition of the pore-forming activity of GSDMB in a liposome leakage assay. The initial slope of fluorescence increase (RU min$^{-1}$) was plotted against the logarithm of IpaH7.8 concentration to determine the IC$_{50}$ of IpaH7.8. RU, relative unit of fluorescence. **a,b**, Each dot represents mean ± s.d. of three technical replicates. **c**, Inhibition of GSDMB-mediated bacterial killing by IpaH7.8. WT: wild type; YY: Y165E/Y166R; RR: R186E/R228D. Error bars, mean ± s.d. of three independent experiments. *$P < 0.05$, **$P < 0.005$, ***$P < 0.0005$. One-way analysis of variance (ANOVA) followed by Tukey's post hoc test compared with buffer control. **d**, Liposome sedimentation assay showing the association of ubiquitinated and non-ubiquitinated GSDMB with CL-liposomes. Supernatants and pellets (liposomes) were analysed by SDS–PAGE stained with Coomassie blue. FL, full length. **e**, Effect of ubiquitination on inhibition of pore-forming activities of GSDMB in induction of fluorescence release from CL-liposomes. Each dot represents mean ± s.d. of three technical replicates. **f**, Lysines at the transmembrane region, lipid-binding and oligomerization interfaces in both GSDMB and human GSDMD are shown. **g**, Coomassie blue-stained SDS–PAGE of in vitro ubiquitination of GSDMB mutants harbouring no lysine (allKR) or single lysine (as indicated). Results representative of three independent experiments. **h**, Monitoring of pore-forming activities of ubiquitinated GSDMB mutants from **g** using with CL-liposomes. Fluorescence measurements were taken at the time point of 30 min. Results represent mean ± s.d. of three technical replicates. **i**, Monitoring the pore-forming activity of ubiquitinated GSDMB$^{3R}$ mutant. GSDMB$^{3R}$: K177/K190/K192 are mutated into arginines in GSDMB. Each dot represents mean ± s.d. of three technical replicates.

was significantly rescued by the addition of wild-type (WT) IpaH7.8 but not by mutants IpaH7.8$^{Y165E/Y166E}$ and IpaH7.8$^{R186E/R228D}$. Full-length GSDMB or IpaH7.8 alone showed no effect on bacterial growth (Fig. 3c). These observations suggest that IpaH7.8 is a direct inhibitor of GSDMB.

## Direct inhibition by ubiquitination

IpaH7.8-mediated ubiquitination itself was insufficient to prevent GSDMD from associating with membranes[18] (Extended Data Fig. 5f). The inhibition of GSDMD-mediated pyroptosis by IpaH7.8 relies on subsequent proteasomal degradation[18]. Consistent with this report, our liposome leakage assay showed an activity of ubiquitinated human GSDMD comparable to non-ubiquitinated (Extended Data Fig. 5g). Unexpectedly, however, ubiquitinated GSDMB almost completely lost its ability to associate with liposomes, and its pore-forming activity to induce liposome leakage (Fig. 3d,e).

We then wondered which lysines in GSDMB-N are responsible for ubiquitination-mediated inhibition of pore-forming activity. GSDMB contains 31 lysines with 16 in the pore-forming domain whereas there are only 15 lysines in human GSDMD, among which 12 are in the pore-forming domain. Among these lysines, K14 is unique in GSDMB and localizes on the C terminus of helix α1, a region involved in GSDM pore assembly[9,23], and K43 locates near the lipid-binding interface. Ubiquitination of K14 and K43 may affect GSDMB-N oligomerization and lipid binding. K102, K177, K190 and K192 locate in the transmembrane region and are conserved in both GSDMB and GSDMD; however, they are buried further in the membrane in GSDMB than in GSDMD

(Fig. 3f and Extended Data Fig. 6). Ubiquitination of these residues may affect the membrane insertion of GSDMB. To examine these possibilities experimentally, we generated a GSDMB construct with all lysines mutated into arginines (mutant allR), and constructs possessing either a single lysine (K14, K102 or K177) or two lysines (K190/K192). In vitro ubiquitination assay showed that all these GSDMB constructs triggered the activation of E3 ligase activity of IpaH7.8, indicating the correct folding of these mutants and their ability to bind IpaH7.8 (ref. [24]) (Fig. 3g, poly-Ub). However, only GSDMB$^{K177}$ and GSDMB$^{K190/K192}$ were ubiquitinated by IpaH7.8 (Fig. 3g, disappearance of GSDMB-FL compared with inputs). Correspondingly, the ubiquitinated GSDMB$^{K177}$ and GSDMB$^{K190/K192}$ mutants showed significantly attenuated activities in induction of liposome leakage (Fig. 3h). Next, we mutated all three lysines (K177, K190 and K192) in WT GSDMB (mutant 3R) and tested the effect on ubiquitination and pore-forming activity. The GSDMB$^{3R}$ mutant still underwent ubiquitination by IpaH7.8, indicating the existence of other ubiquitination sites in GSDMB[4,18] (Extended Data Fig. 5h). However, ubiquitinated GSDMB$^{3R}$ possessed pore-forming activity comparable to non-ubiquitinated (Fig. 3i). Collectively, we conclude that the ubiquitination of GSMDB by IpaH7.8 at K177, K190 and K192 is sufficient to inhibit its pore-forming activity without the requirement for proteasomal degradation.

## GSDMB isoforms exhibit distinct activity

Despite its direct bactericidal activity[4], it remains inconclusive whether GSDMB induces pyroptosis. We noticed that the interdomain linkers in

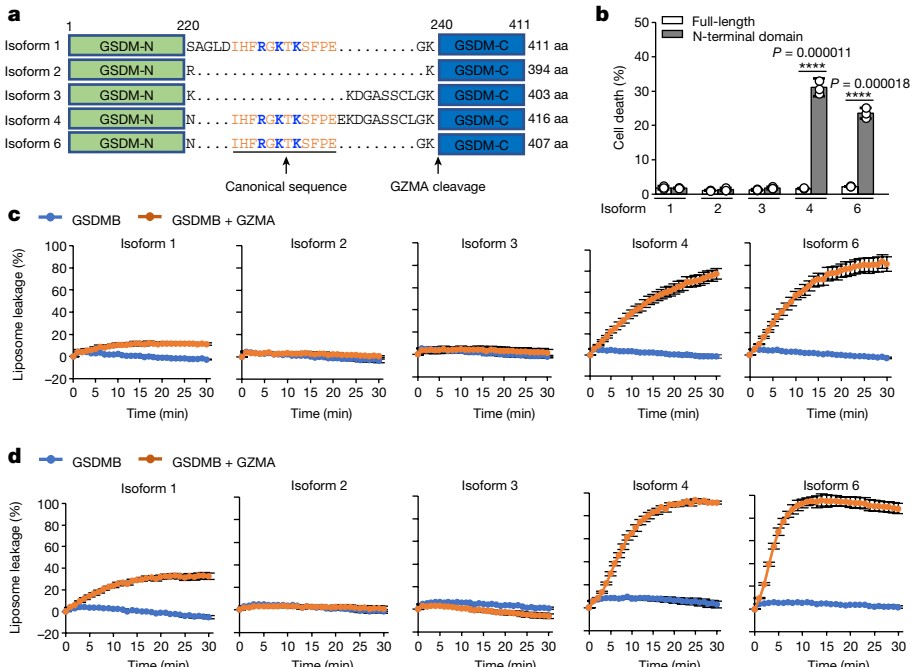

**Fig. 4 | GSDMB isoforms exhibit distinct pore-forming activity both in vitro and in cells. a**, Scheme of GSDMB isoforms 1–4 and 6. Amino acid (aa) sequences of interdomain linkers are shown. Canonical sequences are coloured orange and underlined, the three positively charged residues that mediate lipid binding are highlighted in blue and the GZMA cleavage site is labelled. Isoform 5 is not shown because it does not contain the N-terminal pore-forming domain. **b**, Cytotoxicity of GSDMB isoforms as measured by Hoechst/propidium iodide (PI) double-staining assay in transiently transfected HEK293T cells. Error bars, mean ± s.d. of three independent experiments. One-way ANOVA followed by Tukey's post hoc test. ****$P < 0.00001$. **c,d**, Monitoring of pore-forming activities of purified GSDMB isoforms using liposomes containing 10% PS (**c**) and liposomes comprising live polar lipid extracts (**d**). Each dot represents mean ± s.d. of three technical replicates.

five human GSDMB isoforms (isoforms 1–4 and 6) vary in both length and sequence[25] (Fig. 4a and Extended Data Fig. 6). The linker does not participate in GSDMB–IpaH7.8 interaction (Fig. 1e), and all five GSDMB isoforms were equally targeted by IpaH7.8 for ubiquitination and inhibition (Extended Data Fig. 7a,b); neither does the linker affect GZMA cleavage (Extended Data Fig. 7c). Rather, the linker may play roles in regulation of GSDMB pore-forming activity[3]. Previous studies have shown that GSDMB isoforms 4 and 6 containing the canonical sequence in the interdomain linker showed strong membrane-permeabilizing activities both in vitro and in cells[5,7], whereas isoform 2, which lacks the canonical sequence, does not have pyroptotic activity[3]. Surprisingly, isoform 1, also containing the canonical sequence in its interdomain linker, does not induce pyroptotic cell death; rather, it targets bacterial membranes to kill bacteria[4]. Consistently, we found that cells transfected with the N-terminal domain of isoforms 4 or 6, but not that of isoforms 1, 2 or 3, exhibited marked pyroptotic cell death as compared with cells transfected with full-length GSDMBs (Fig. 4b and Extended Data Fig. 7d). Interestingly, isoform 4-mediated pyroptotic cell death was significantly inhibited when coexpressed with *Shigella* IpaH7.8. This inhibition was independent of the E3 ligase activity of IpaH7.8 (Extended Data Fig. 7e,f), further confirming that IpaH7.8 binding had directly inhibited the pore-forming activity of GSDMB. To further examine the pore-forming activities of GSDMB isoforms against mammalian plasma membranes in vitro, we performed liposome leakage assay using liposomes containing 10% phosphatidylserine (PS). Our results showed that isoforms 4 and 6 exhibited strong permeabilizing activities towards PS-liposomes whereas isoforms 2 and 3 did not show membrane-permeabilizing activity (Fig. 4c). Similar results were also observed for liposomes comprising liver polar lipid extracts (Fig. 4d). Interestingly, isoform 1 exhibited only 20–40% of pore-forming activity as compared with isoforms 4 and 6 (Fig. 4c,d). The weak permeabilizing activity of GSDMB isoform 1 is probably insufficiently strong to

overcome membrane repair efforts by cellular machinery, including ESCRT-III[26], to induce pyroptosis, whereas it is sufficient to kill bacteria lacking membrane repair mechanisms, although the toxicity of isoform 1 to bacteria was slightly weaker than that of isoforms 4 and 6 (Extended Data Fig. 7g). These data indicate that GSDMB isoforms varying in their interdomain linkers exhibit distinct pore-forming activities.

## Cryo-EM structure of GSDMB pore

To address the question of why GSDMB isoforms exhibit distinct pore-forming activities, we sought to determine the cryo-EM structure of the GSDMB pore. Although GSDMB isoforms 1, 4 and 6 formed pores similar in size and shape, those of isoform 1 showed less aggregation than those of isoforms 4 and 6, and isoform 1 pores were seen to be distributed evenly on cryo-EM grids (Extended Data Fig. 8a,b). We thus subjected GSDMB isoform 1 to cryo-EM analysis. Three-dimensional (3D) classification of the collected cryo-EM dataset yielded a major class of GSDMB β-barrel pores, a class of rings without a β-barrel representing prepores and other classes representing GSDMB prepore–pore transition intermediate states[9,23] (Extended Data Figs. 8c and 9a,b). The GSDMB pore was found to be 24–26-fold symmetric (Fig. 5a). 3D refinement of the 24-fold symmetric pore led to a final map at 4.96 Å overall resolution whereas focus refinement improved local resolution of the globular domain to 4.48 Å (Extended Data Fig. 8c,d and Extended Data Table 1), allowing us to build an atomic model of the GSDMB pore using the structure of GSDMB in the complex of GSDMB/IpaH7.8 as a starting model.

The 24-fold GSDMB pore has an estimated inner diameter of 150 Å, outer diameter of 250 Å and height of 60 Å (Fig. 5a), very similar to the 27-fold GSDMA3 pore but much smaller than the 31-fold GSDMD pore[9,23]. Similar to that in GSDMA3 and GSDMD, each GSDMB pore subunit comprises a globular domain ('palm') and two inserted β-hairpins

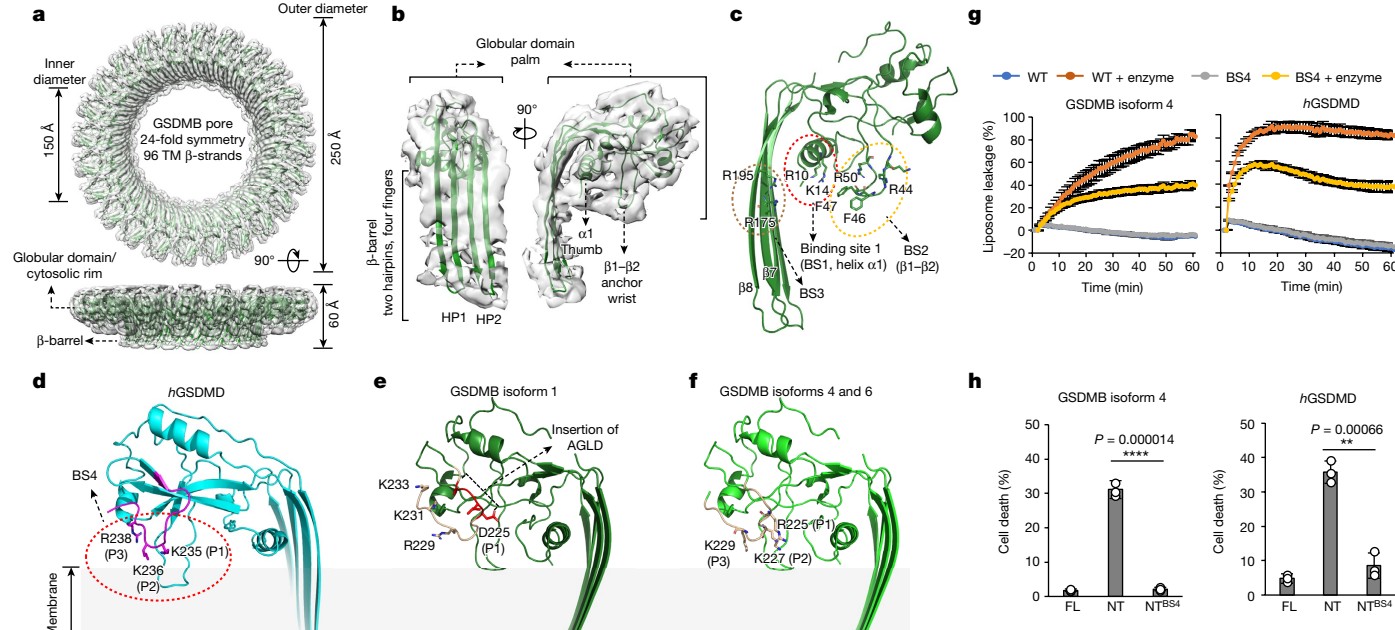

**Fig. 5 | Cryo-EM structure of GSDMB pore identifies the interdomain linker as a regulator of the pore-forming activity of GSDMB isoforms. a**, Ribbon diagram of the 24-subunit GSDMB isoform 1 pore structure fitted to its cryo-EM density map. TM, transmembrane. **b**, Structure of a GSDMB subunit in its pore conformation fitted to the cryo-EM map. **c**, Three lipid-binding sites (BS1–3) in GSDMB are shown, located, respectively, at α1 and loops β1–β2 and β7–β8. Residues involved in lipid binding are labelled. **d**–**f**, Diagram showing how human GSDMD (**d**), GSDMB isoform 1 (**e**) and GSDMB isoforms 4 and 6 (**f**) anchor on membranes. Interdomain linkers in GSDMs are highlighted by the colours shown. Positively charged residues labelled in the interdomain linker form an extra lipid-binding site (BS4). GSDMB isoform 1 preserves only one positively charged residue in the interdomain linker because of a four-amino-acid insertion.

**g**, Monitoring the pore-forming activities of GSDMB isoform 4 (left) and human GSDMD (hGSDMD, right) by liposome leakage assay. BS4: triple mutations of the three basic residues (R225/K227/K229 in GSDMB isoform 4 and K235/K236/R238 in hGSDMD) to glutamic acids. GSDMB isoform 4 and hGSDMD were cleaved by GZMA and active caspase-11, respectively. Each dot represents mean ± s.d. of three technical replicates. **h**, Effect of BS4 of GSDMB isoform 4 and hGSDMD on induction of pyroptotic cell death in HEK293T cells as monitored by Hoechst/PI double-staining assay. NT, N-terminal domain; NT[BS4], N-terminal domain harbouring BS4 mutation. Error bars, mean ± s.d. of three independent experiments. One-way ANOVA followed by Tukey's post hoc test, **P < 0.005, ****P < 0.00005.

('fingers') generated from residues in extension domains 1 and 2 (ED1 and ED2, respectively) in the full-length auto-inhibited GSDMB (Fig. 5b and Extended Data Fig. 9c). Analysis of the GSDMB pore showed conserved oligomerization interfaces previously observed in both GSDMA3 and GSDMD[9,23]. GSDMB pore oligomerization is mediated by both the inserted β-strands in the transmembrane region and the cytosolic globular domains (Extended Data Fig. 9d). The interaction in the transmembrane region is contributed by residues running along the neighbouring β3 and β8 strands between the subunits (Extended Data Fig. 9d). The interaction in the adjacent globular domains contains mostly residues from helix α3 of one subunit interacting with the region around α2 and β11 of its neighbouring subunit (Extended Data Fig. 9d); and the α1 helix from one subunit juxtaposing end-on with the α1 helix from the next subunit through hydrogen-bonding and hydrophobic interactions (Extended Data Fig. 9d). These conserved interactions suggest a unified oligomerization mechanism in the GSDM family, despite their variability in the assembly stoichiometry.

## The linker regulates GSDMB activity

Previous studies have identified the N-terminal α1 helix ('thumb', binding site 1 (BS1)), the β1–β2 loop with a hydrophobic tip flanked by positively charged residues ('wrist', binding site 2 (BS2)) in the GSDM globular domain and a positively charged lipid-binding site (binding site 3 (BS3)) present on the membrane-inserted β7–β8 hairpin as structural elements for lipid binding[8,9,23]. As expected, all three binding sites are conserved in GSDMB (Fig. 5c). BS1 and BS2 contain basic residues of R10 and K14 in the α1 helix and K43, R44 and R50 in the β1–β2 loop interacting with the acidic lipid head groups, and hydrophobic residues

of F46 and F47 in the β1–β2 loop hydrophobic tip inserting into the lipid bilayer as a membrane anchor, whereas BS3 is mediated by basic residues of R174 and R195 in β7 and β8 (Fig. 5c).

We then examined the interdomain linker. In the human GSDMD pore, the density of the entire interdomain linker (V229–Q241) is visible (Extended Data Fig. 9e). The first few residues of the interdomain linker in GSDMD (region 1: V229–F232) are required for pore oligomerization[8] (Extended Data Figs. 6 and 9e). Mutation of the corresponding region in GSDMB isoforms 1 and 4 ('AGLD' in isoform 1 and 'NIHF' in isoform 4 to 'GGGG', respectively) markedly compromised their pore-forming activities (Extended Data Fig. 9g,h), indicating a similar role of region 1 in GSDMB. The following sequence (region 2: P233–Q241 in human GSDMD) in the interdomain linker was not considered a structural element involved in pore formation previously[8,9,23]. Surprisingly, the density of the region 2 in GSDMB pore is also visible, regardless of the relatively lower resolution (Extended Data Fig. 9f). We suggested that region 2 may be stabilized by certain interactions in the pore. Region 2 in human GSDMD contains three basic residues with their positively charged side chains pointing toward the membrane, probably forming an extra lipid-binding site (BS4) for membrane attachment (Fig. 5d). The canonical interdomain linker in GSDMB also contains basic residues in region 2 (Fig. 4a and Extended Data Fig. 6). However, only one basic residue (R229) is structurally conserved in the GSDMB isoform 1 pore. Due to a four-amino-acid (_222_AGLD_225_) insertion in the interdomain linker, the residue at the first position (P1) is replaced by a negatively charged D225 (Fig. 5e). Substitution by an acidic residue at this position probably weakens membrane attachment by repelling the acidic membrane surface, thus attenuating the pore-forming activity of GSDMB isoform 1. The generated atomic models of other GSDMB

isoforms covering entire interdomain linkers, based on the structure of isoform 1 and with the assistance of AlphaFold prediction[27], show that isoforms 4 and 6 are structurally conserved to human GSDMD and preserve all three basic residues, R225, K227 and K229 (Fig. 5f). By contrast, isoform 3 with a truncated interdomain linker would probably preserve the oligomerization interface but lacks the basic cluster for lipid binding, and isoform 2 lacks the entire interdomain linker for oligomerization and lipid binding. A recent study showing that GSDMB isoforms without exon 6, which encodes the canonical sequence in the interdomain linker, did not induce pyroptosis strongly supports our model[3].

Triple mutation of R225E/K227E/K229E in the interdomain linker of GSDMB isoform 4 significantly compromised its activity to induce liposome leakage in vitro (Fig. 5g), and to mediate pyroptotic cell death (Fig. 5h and Extended Data Fig. 9i), confirming the critical role of BS4 in mediating GSDMB pore formation. Similar results were also observed when we mutated the corresponding residues in human GSDMD (Fig. 5g,h and Extended Data Fig. 9j). Collectively, we conclude that the interdomain linker is the key structural element regulating the pyroptotic activity of GSDMB isoforms, by mediation of pore oligomerization and provision of an extra lipid-binding site.

## Discussion

Our cryo-EM structures of the GSDMB–IpaH7.8 complex and GSDMB pore demonstrate the structural mechanisms underlying GSDMB recognition by the bacterial effector and the pyroptotic activity of GSDMB, respectively.

The structure of the GSDMB–IpaH7.8 complex identifies a motif of three negatively charged residues in the N-terminal α1–β1′ loop in GSDMB and human GSDMD as the structural determinant specifically recognized by *Shigella* IpaH7.8. IpaH7.8 does not bind mouse GSDMD, in which the α1–β1′ motif is not conserved, leading to the inability of *Shigella* to efficiently establish infection in the mouse. Previously, IpaH7.8 was reported to bind mouse GSDMD at an even stronger affinity than human GSDMD[18], which is inconsistent with our results. This discrepancy can be attributed to the different methods used. The microscale thermophoresis method used in that previous study requires labelling of the protein with hydrophobic fluorescent dyes, which may change protein behaviour, leading to confounding results[28,29]. Alternatively, the ITC method we used here reliably measures the binding affinity of proteins in their native states without the requirement for labelling[30]. This is also supported by our mutagenesis experiments and a recent study report[31].

Our study demonstrates highly efficient inhibition of GSDMB by IpaH7.8: binding of IpaH7.8 to GSDMB directly prevented its association with the membrane. Moreover, the subsequent ubiquitination of GSDMB by IpaH7.8, when *Shigella* hijacks the host ubiquitination system, further inhibits GSDMB activity. This inhibition is mediated by the ubiquitination of three lysines in the second transmembrane hairpin of GSDMB. Ubiquitination probably affects the membrane insertion of GSDMB, thus inhibiting its pore-forming activity. This multipronged inhibition of GSDMB by IpaH7.8 endows *Shigella* with a very efficient way to escape from attack by cytotoxic lymphocytes and natural killer cells during infection[5], promoting bacterial survival in the host replicative niche.

The cryo-EM structure of the GSDMB pore illustrates a unique interdomain linker-regulated mechanism of pore oligomerization and lipid binding. GSDMB is widely expressed in various cell types and tissues[5,32,33], where its isoforms with distinct interdomain linkers may be differentially regulated. Such differences in the expression and activities of GSDMB isoforms probably represent a host strategy to fine-tune cell type-specific outcomes of GSDMB activation. For example, pyroptotic GSDMB isoforms are dominant in epithelial cells and are responsible for elimination of the replicative niche of intracellular pathogens through induction of pyroptosis in infected cells, whereas non-pyroptotic isoform 1 may be dominant in macrophages or dendritic cells where it targets and kills cytosolic bacteria—rather than inducing pyroptosis—thus ensuring the survival of these antigen-presenting cells for T cell activation. Currently the cell-specific distribution, abundance and function of each GSDMB isoform are not well understood. Physiological relevance of GSMDB isoforms to antibacterial immunity needs further investigation.

Both GSDMB and GSDMD are generally important in regard to innate immunity to bacterial pathogens. However, a pathogen such as *Shigella* interfering with GSDMB and GSDMD in humans minimizes their contribution to host defence, explaining why humans are susceptible to *Shigella* whereas mice, which lack GSDMB and whose GSDMD is not sensitive to IpaH7.8, exhibit resistance[4,18]. It's worth noting that people with shigellosis usually recover in 5–7 days without needing antibiotics[34], suggesting that GSDMB and GSDMD, although targeted by IpaH7.8, could still play roles in rendering *Shigella* infectious to humans.

In addition to bacterial infection, GSDMB is associated with various cancers. One recent study indicated that expression of non-pyroptotic isoform 2 was higher than that of other isoforms in patients with breast cancer[3]. Upregulation of isoforms 2 and 3 may promote tumorigenesis and metastasis, leading to poor overall patient survival, whereas high expression of pyroptotic isoform 4 has the opposite effect[3]. Considering this potential association between GSDMB isoforms and cancer survival, further studies are warranted to explore whether cancer cells exploit differential expression of GSDMB isoforms to resist attack by cytotoxic lymphocytes.

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

## Methods

### Constructs and mutagenesis

The coding sequences of full-length GSDMs were cloned into a pET28-His-SUMO vector after the N-terminal $His_6$-SUMO tag. In regard to the GSDMB construct used for cryo-EM structural determination, a human rhinovirus 3C protease (3C) site (LEVLFQ/GP) was inserted after residue K239. The coding sequence of *S. flexneri* IpaH7.8 was cloned into a pET26b vector with a C-terminal 6XHis tag, and cloned into pET22b vector without an affinity tag for coexpression with GSDMB. Caspase-11 (96–373) was cloned into a pET22b vector to purify the active form of the p20–p10 complex. For the cellular experiments, GSDMs (full length) and GSDM-N were cloned into a pcDNA3.1 vector in which a FLAG-tag was fused at the C terminus while IpaH7.8 was inserted into a pCMV-HA vector, resulting in a fusion protein with an N-terminal HA tag. All mutations in this study were introduced using either the QuikChange Site-Directed Mutagenesis Kit (Stratagene) or Gibson Assembly Master Mix (New England BioLabs), and all plasmids were verified by sequencing.

### Protein expression and purification

To obtain the GSDMB–IpaH7.8 complex, *E. coli* BL21 (DE3) cells harbouring the expression plasmids of pET28-His-SUMO-GSDMB and pET22b-IpaH7.8 were grown in lysogeny broth medium supplemented with 50 μg ml$^{-1}$ kanamycin and 100 μg ml$^{-1}$ ampicillin at 37 °C. Protein expression was induced by the addition of 0.5 mM isopropyl β-D-1-thiogalactopyranoside at 20 °C for 16 h when optical density ($OD_{600}$) reached 0.8. Cells were collected by centrifugation at 5,000$g$ for 20 min. Harvested cells were lysed by sonication in a buffer containing 25 mM Tris-HCl pH 8.0, 150 mM NaCl, 2 mM β-mercaptoethanol and 25 mM imidazole. Lysates were centrifuged at 18,000$g$ and 4 °C for 30 min to remove insoluble fractions. Supernatants containing recombinant proteins were purified using Ni-NTA agarose (Qiagen) according to the manufacturer's instructions. $His_6$-SUMO tag removal was performed on a Ni-NTA column at 4 °C overnight with the addition of the recombinant Ulp1. Flowthrough non-tagged proteins were further purified using a Hitrap Q HP ion-exchange column (Cytiva), then a Superdex Increase 200 (10/300) size-exclusion column (Cytiva) in a buffer containing 25 mM HEPES pH 7.5 and 150 mM NaCl. All purified proteins were confirmed by Coomassie blue staining of SDS–PAGE.

Similar protocols were applied for the expression and purification of all individual GSDMs, IpaH7.8 and their mutants, except that the $His_6$-SUMO tag was retained for GSDMs used in the in vitro ubiquitination assay. All purified proteins were concentrated to approximately 5–10 mg ml$^{-1}$ before use.

The GZMA plasmid pET26b-GZMA was a kind gift from J. Lieberman[35]. The plasmids of E1 (pET21d-hUbE1), E2 (pET15-hUbE2D2) and ubiquitin (pET15-Ub) were kind gifts from C. Wolberger, W. Harper and R. Klevit, respectively[36–38]. Expression and purification followed previous protocols.

### In vitro ubiquitination activity assay

In vitro ubiquitination reactions were performed in buffer A (25 mM HEPES pH 7.5, 50 mM NaCl, 0.1 mM DTT, 10 mM $MgCl_2$ and 5 mM ATP). Components were mixed as indicated at concentrations of 0.4 μM E1 (human UbE1), 2 μM E2 (human UbE2D2), 10 μM E3 (IpaH7.8$^{WT}$ or IpaH7.8$^{C357A}$ mutant), 200 μM ubiquitin and 10 μM GSDM (WT or indicated mutants). Reactions were incubated at 37 °C for 2 h and stopped by the addition of SDS–PAGE loading dye, followed by boiling for 5 min before electrophoresis. Ubiquitination was evaluated by Coomassie blue staining of SDS–PAGE.

### Liposome leakage assay

The liposome leakage assay was performed following an established protocol[9]. Briefly, 1-palmitoyl-2-oleoyl-sn-glycero-3-phosphocholine and PS or CL (Avanti Polar Lipids) were mixed at the indicated ratio in a glass tube. The solvent chloroform was evaporated under a stream of nitrogen gas for 30 min. The dry lipid film was then rehydrated with buffer B (25 mM HEPES pH 7.5, 150 mM NaCl) supplemented with 50 mM 6-FAM (Tokyo Chemical Industry). 6-FAM-loaded liposomes were then extruded through a 1 μm membrane (Whatman Nuclepore) using a mini-extruder (Avanti Polar Lipids). To remove unencapsulated 6-FAM, extruded liposomes were subjected to a PD-10 desalting column (Cytiva) equilibrated with buffer B. For the liposome leakage assay, liposomes were incubated with proteins of GSDMB/D and/or IpaH7.8 with or without activating enzymes (GZMA for GSDMB and caspase-11$^{p10/p20}$ for GSDMD). Reactions were performed on a 384-well plate, with release of 6-FAM dye monitored by fluorescence at 517 nm using a SpectraMax M5 plate reader (Molecular Devices) with excitation at 495 nm for 60 min at 1 min intervals.

### Liposome pulldown assay

Liposomes were prepared as described above, except that fluorescent dye was not used. Liposomes were incubated with GSDMB/D proteins in/without the presence of IpaH7.8 at various molar ratios with or without activating enzymes. Mixtures were incubated for 30 min at 4 °C before sedimentation at 20,000$g$ for 30 min at 4 °C. Supernatants were transferred immediately to new tubes and pellets were washed twice with buffer B, then resuspended in an equal volume of buffer. Proteins in both pellets and supernatant were then analysed by Coomassie blue staining of SDS–PAGE.

### ITC assay

Protein concentrations of non-tagged GSDMs and IpaH7.8 were measured in triplicate using a NanoDrop One Microvolume UV-Vis Spectrophotometer (Thermo Fisher Scientific) based on their extinction coefficients. Isothermal titration calorimetry measurements were performed at 20 °C using a VP-ITC microcalorimeter (MicroCal). Experiments were performed by the injection of 250 μl of IpaH7.8 solution (200 μM) into a sample cell containing 2 ml of GSDMB (10 μM) in 25 mM Tris-HCl pH 8.0 and 150 mM NaCl. In total, 25 injections were administered at 300 s intervals. In regard to human GSDMD (*h*GSDMD) and mouse GSDMD (*m*GSDMD), 250 μl of IpaH7.8 solution (625 μM) was titrated into a sample cell containing 2 ml of either *h*GSDMD (40 μM) or *m*GSDMD (40 μM). All ITC data were analysed using Origin Software provided by the manufacturer and fitted to a one-site binding model.

### Cell culture and transfection

The 293T cells were obtained from the American Type Culture Collection and were frequently checked in regard to their morphological features and functionalities. Cells were grown in DMEM (Gibco) supplemented with 10% (v/v) fetal bovine serum (Gibco) and 2 mM L-glutamine at 37 °C in a 5% $CO_2$ incubator. Transient transfection in 293T cells was performed using Lipofectamine 3000 (Thermo Fisher Scientific) following the manufacturers' instructions.

### Immunoprecipitation assays

For detection of GSDMB ubiquitination in cells, pcDNA-FLAG-GSDMB (isoform 1) was cotransfected with pCMV-HA-IpaH7.8 (WT or indicated mutants) into HEK293T cells in a 10 cm tissue culture dish. Eight hours after transfection, a final concentration of 10 μM bortezomib (Sigma Aldrich) was added to the cell culture to reduce proteasome-mediated protein degradation. After a further 8 h, cells were collected and lysed in 25 mM Tris-HCl pH 7.5, 150 mM NaCl, 0.5% NP-40 and 1× protease inhibitor cocktail (Sigma Aldrich). Lysate was added to 25 μl of Anti-FLAG M2 Magnetic Beads (Sigma Aldrich, no. M8823) and incubated at 4 °C for 3 h with gentle rotation. Beads were washed three times with PBS buffer then eluted with 50 μl of PBS buffer containing 100 μg ml$^{-1}$ FLAG peptide (Sigma Aldrich, no. F3290). Eluted samples were boiled with an equal volume of 2× SDS Loading buffer (Bio-Rad) then processed for

immunoblotting with one of the following antibodies: anti-FLAG (Sigma Aldrich, no. F1804, 1:1,000), anti-actin (Cell Signaling Technology, no. 3700S, 1:1,000), anti-HA (Cell Signaling Technology, no. 3724S, 1:1,000) or anti-ubiquitin (Thermo Fisher Scientific, no. PA3-16717, 1:1,000).

### Cellular degradation assay
One each of plasmids pcDNA-FLAG-GSDMB and pcDNA-FLAG-GSDMD (250 ng) (WT or indicated mutants) was cotransfected with 500 ng of the pCMV-HA-IpaH7.8 plasmid into HEK293T cells seeded in a 12-well plate at $1.5 \times 10^5$ cells per well. After 40 h cells were lysed in RIPA buffer (Thermo Fisher Scientific), added to an equal volume of 2× SDS Loading buffer (Bio-Rad) and processed for immunoblotting.

### Cytotoxicity assay
Cell death was determined by Hoechst/PI double-staining assay: 150 ng of the indicated pcDNA-FLAG-GSDMB construct or 75 ng of pcDNA-FLAG-GSDMD plasmid (FL, NT or indicated mutants) was transfected into HEK293T cells seeded in a 96-well plate at $2 \times 10^4$ cells per well. For IpaH7.8 inhibition, pcDNA-FLAG-GSDMB-NT (isoform 4) was cotransfected with 200 ng of the pCMV-HA-IpaH7.8 plasmid (WT or C357A). Transfected cells were then cultured for up to 40 h. At the start of the assay, cells were stained with 30 μM PI (Sigma Aldrich) for 10 min followed by 15 μM Hoechst 33342 (Thermo Fisher Scientific) for 15 min at 37 °C in the dark. Afterwards, cells were visualized using a ZOE Fluorescent Cell Imager (Bio-Rad). Cell death was quantified and expressed as the percentage of PI-positive cells among total cells (Hoechst-stained cells).

### Bacterial growth inhibition assay
*Escherichia coli* DH5α was grown overnight in BHI medium, then diluted the following day at 1:100 in BHI and grown for a further 2 h at 37 °C until exponential phase. Next, 1 ml of the bacterial culture was collected with centrifugation at 5,000$g$ for 2 min and resuspended in buffer B to a final bacterial cell density of $5 \times 10^8$ ml$^{-1}$. For the killing assay, 5 μl of bacteria was added to a 15 μl reaction containing 10 μM full-length GSDMB in the absence or presence of GZMA. Reactions were performed at 37 °C for 2 h. After incubation, 5 μl of treated bacteria was seeded into 200 μl of BHI in flat-bottomed, 96-well plates. Bacterial growth was monitored by reading absorbance at 600 nm over 6 h using a SpectraMax M5 plate reader (Molecular Devices). Numbers of recovered colony-forming units (CFUs) were calculated by normalization of the OD$_{600}$ of treated bacteria (with GZMA in reactions) to untreated bacteria (buffer or without GZMA).

### GSDMB pore reconstitution and purification
Purified GSDMB isoform 1 was added to the prepared liposomes, followed by the addition of 3C protease to initiate pore formation. The reaction proceeded on ice for 3 h. Liposomes loaded with GSDMB pore were solubilized by 2% C12E8 (Anatrace) to extract pores. To remove poorly behaving particles and GSDMB-C, samples were further purified using a Superose 6 (10/300) Increase size-exclusion column (Cytiva) equilibrated with buffer B (25 mM HEPES pH 7.5, 150 mM NaCl and 0.006% C12E8).

### Negative-staining electron microscopy
For negative staining, 10 μl of the GSDMB–IpaH7.8 complex or GSDMB pore was applied to a glow-discharged, carbon-coated copper grid (Electron Microscopy Sciences). The sample was incubated on the grid for 1 min, stained with 1% uranyl acetate for 1 min and blotted dry. Grids were imaged on a Hitachi H-7650 transmission electron microscope equipped with a 2k CCD camera (Advanced Microscopy Techniques) at the UCONN Health Electron Microscopy Facility.

### Cryo-EM grid preparation and data acquisition
For the GSDMB–IpaH7.8 complex, 3.5 μl of freshly purified sample at 0.5 mg ml$^{-1}$ was applied to plasma glow-discharged, Quantifoil holey copper grids (R 1.2/1.3, 400 mesh, Electron Microscopy Sciences) using a Vitrobot Mark IV (Thermo Fisher Scientific) set at blotting force 4, blotting time 5.5 s, 100% humidity and 4 °C. Blotted grids were immediately plunged into liquid ethane and transferred to liquid nitrogen for storage. One cryo-EM dataset was collected at the Case Western Reserve University cryo-EM facility on a Titan Krios electron microscope (Thermo Fisher Scientific) equipped with a K3 Summit direct electron detector (Gatan) and a post-column energy filter (Gatan) in counting mode using serialEM. A total of 3,128 movies were recorded at defocus values ranging from −0.8 to −2.5 μm at magnification ×105,000 and pixel size 0.414 Å. For each movie, 58 frames were acquired over 5.25 s at an approximate total dose of 66.95 e$^-$ Å$^{-2}$.

For GSDMB pores, detergent-solubilized GSDMB pores were concentrated to 0.6 mg ml$^{-1}$ then frozen onto Quantifoil holey copper grids coated with ultrathin carbon film (R 1.2/1.3, 400 mesh, Electron Microscopy Sciences). Briefly, a 3 μl drop of GSDMB pore sample was applied to a plasma glow-discharged lacey carbon grid mounted on a Vitrobot. The grid was then blotted with filter paper for 6 s at blotting force 10 after a waiting time of 2 s. Humidity and temperature in the Vitrobot were set to 100% and 4 °C, respectively, throughout the operation. The blotted grid was then plunged into liquid ethane and transferred to liquid nitrogen for storage. The cryo-EM dataset was collected at the cryo-EM facility at the University of Massachusetts Chan Medical School on a Titan Krios electron microscope (Thermo Fisher Scientific) equipped with a K3 Summit direct electron detector (Gatan) and a post-column energy filter (Gatan). A total of 6,376 movies were collected in counting mode, each containing 50 frames and a total exposure dose of 50 e$^{-1}$ Å$^{-2}$. Magnification was set to 105,000, pixel size was 0.83 Å and defocus range −1.0 to −2.0 μm.

### Cryo-EM image processing
Raw movies were corrected by gain reference and for beam-induced motion and summed into motion-corrected images using MotionCor2 (ref. [39]). CTF parameters were determined using CTFFind4 (ref. [40]) and refined later in cryoSPARC[41].

For the GSDMB–IpaH7.8 complex, after particle picking using the general model in crYOLO[42] the coordinates (1,522,742 particles in total) were transferred to cryoSPARC for subsequent processing. Several rounds of 2D classification were performed to eliminate ice, carbon edges and false-positive particles containing noise. Frequently featured classes containing 307,276 particles were selected and subjected to ab initio 3D reconstruction followed by heterogeneous refinement. The optimal class, containing 113,959 particles, was selected for homogenous and non-uniform refinement[43]. Resolution of the final electron density map was estimated at 3.8 Å, based on the gold-standard Fourier shell correlation (FSC) criterion of 0.143 (ref. [44]). The local resolution distribution of the map was determined by ResMap[45]. The density map sharpened in cryoSPARC was used to produce figures.

For GSDMB pores, a total of 692,212 particles were initially extracted by both manual and automated particle picking in cryoSPARC. Two-dimensional classification was performed in cryoSPARC to eliminate ice, carbon edges and false-positive particles containing noise. After 2D classification, 156,037 particles were imported into Relion-4.0 for 3D classification with an initial model generated de novo in cryoSPARC using the same particle set. C1 symmetry was used for the first round of 3D classification; 3D classes with relatively clear features of C24 symmetry were selected for an extra round of 3D classification with C24 symmetry to discard bad particles. Next, 41,799 particles from the 3D class with optimal resolution were imported back to cryoSPARC for non-uniform refinement. With C24 symmetry, the resolution of the GSDMB pore map was 4.96 Å as measured by gold-standard FSC of 0.143. Focus refinement with a mask excluding the β-barrel region improved local resolution of the GSDMB globular domain to 4.48 Å.

### Model building and structure analysis
Atomic models of both the IpaH7.8–GSDMB complex and GSDMB pore were built and refined into cryo-EM density using Coot[46] and PHENIX[47].

For the IpaH7.8–GSDMB complex, AlphaFold2-predicted structures of IpaH7.8 and GSDMB were used as starting models[27]. Models of IpaH7.8 and GSDMB were docked into EM density as a rigid body in UCSF Chimera[48] then manually adjusted in Coot. The structural model of the complex was further refined using 'phenix.real_space_refine', with secondary structure restraints and Coot iteratively. The quality of the atomic model was evaluated by Molprobity[49]. For GSDMB pores, the structure of GSDMB in the IpaH7.8–GSDMB complex was used as a starting model. A similar procedure was then performed for further adjustment and refinement. Figures were prepared using PyMOL (Schrödinger) and UCSF Chimera.

## Reporting summary

Further information on research design is available in the Nature Portfolio Reporting Summary linked to this article.

## Data availability

The atomic coordinates of the GSDMB–IpaH7.8 complex, GSDMB pores and GSDMB pores without the β-barrel have been deposited in the Protein Data Bank (PDB) under accession nos. 8EFP, 8ET2 and 8ET1, respectively. The associated cryo-EM density maps have been deposited in the Electron Microscopy Data Bank under accession nos. EMD-28087, EMD-28584 and EMD-28583, respectively. All other data are available from the corresponding author on reasonable request. Several structural coordinates in the PDB database used in this study can be located by accession nos. 6CB8, 5B5R, 6N9O, 6N9N, 6VFE, 7V8H and 3CVR.

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

**Acknowledgements** We thank S. Dou (QuintaraCT, Inc.), X. Li and Y. Wang for discussions. This work was supported by the UConn Health Start-up Fund and the US National Institutes of Health (grant nos. R01AI158435 to J.R. and R01AI119015 to V.A.R.).

**Author contributions** J.R. and C.W. conceived the study. C.W., T.T. and D.M. expressed and purified proteins. C.W. reconstituted GSDMB pores. C.W. screened grids and collected cryo-EM data, assisted by J.C., K.L., K.S. and C.X. C.W. and J.R. determined cryo-EM structures. C.W. and J.R. performed biochemical experiments. S.S., T.T., S.W. and C.W. performed cellular experiments. J.R. and V.A.R. supervised the project. All authors organized and analysed data. J.R. and C.W. wrote the paper, with input from all authors.

**Competing interests** The authors declare no competing interests.

**Additional information**
**Correspondence and requests for materials** should be addressed to Jianbin Ruan.

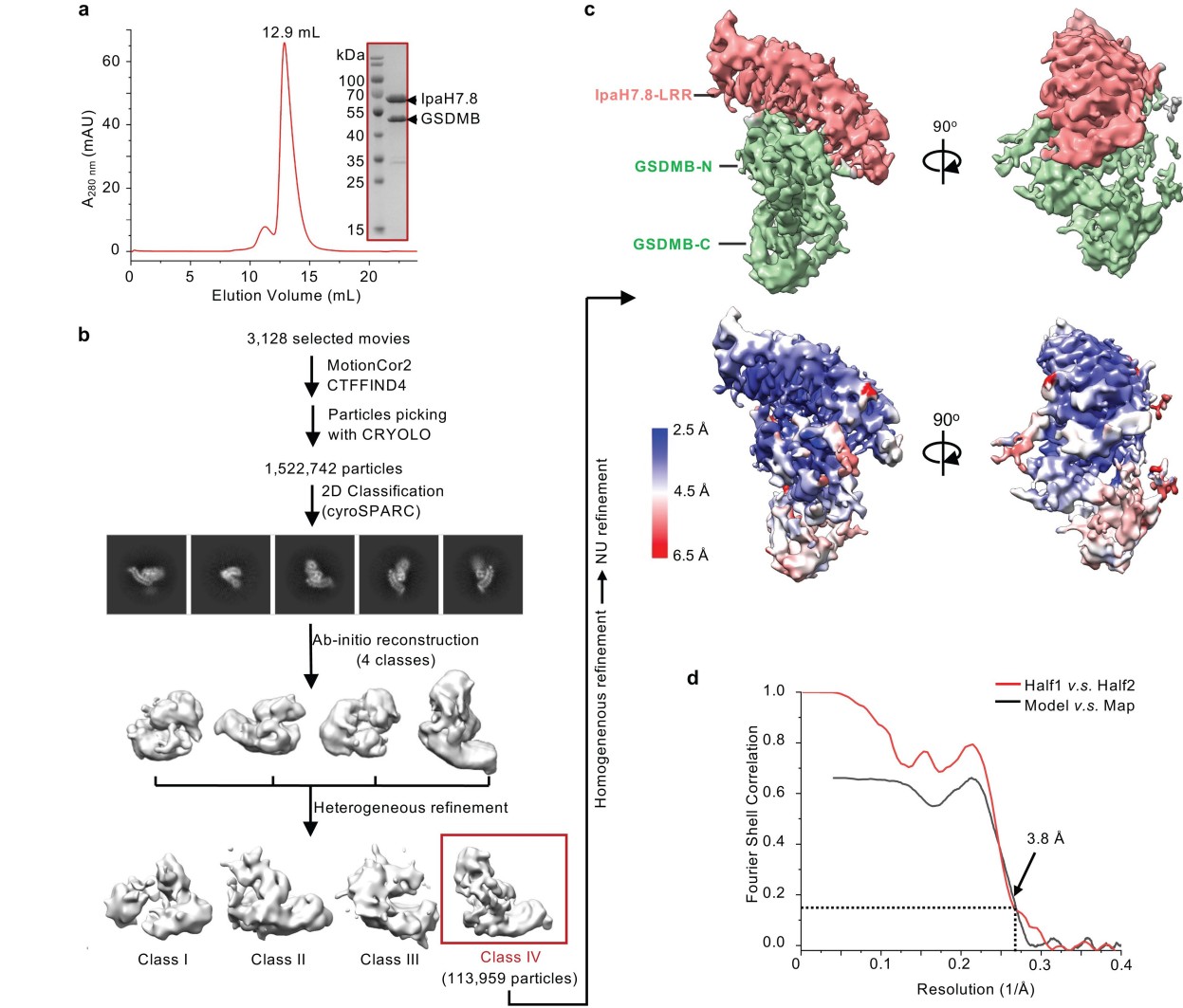

**Extended Data Fig. 1 | Cryo-EM structural determination of GSDMB-IpaH7.8 complex. a**, Gel filtration profile and Coomassie blue-stained SDS-PAGE of the GSDMB-IpaH7.8 complex. Results representative of more than 3 independent experiments. **b**, A brief flow chart of single-particle cryo-EM data collection and process. **c**, Cryo-EM map (Upper panel) and local resolution estimation (lower panel) of the GSDMB-IpaH7.8 complex calculated using ResMap. The highest resolution is observed at the IpaH7.8-LRR domain and GSDMB-N domain. GSDMB-C exhibits relatively low resolution. **d**, The gold-standard Fourier shell correlation (FSC) curve for the overall map and model-to-map correlations of the GSDMB-IpaH7.8 complex.

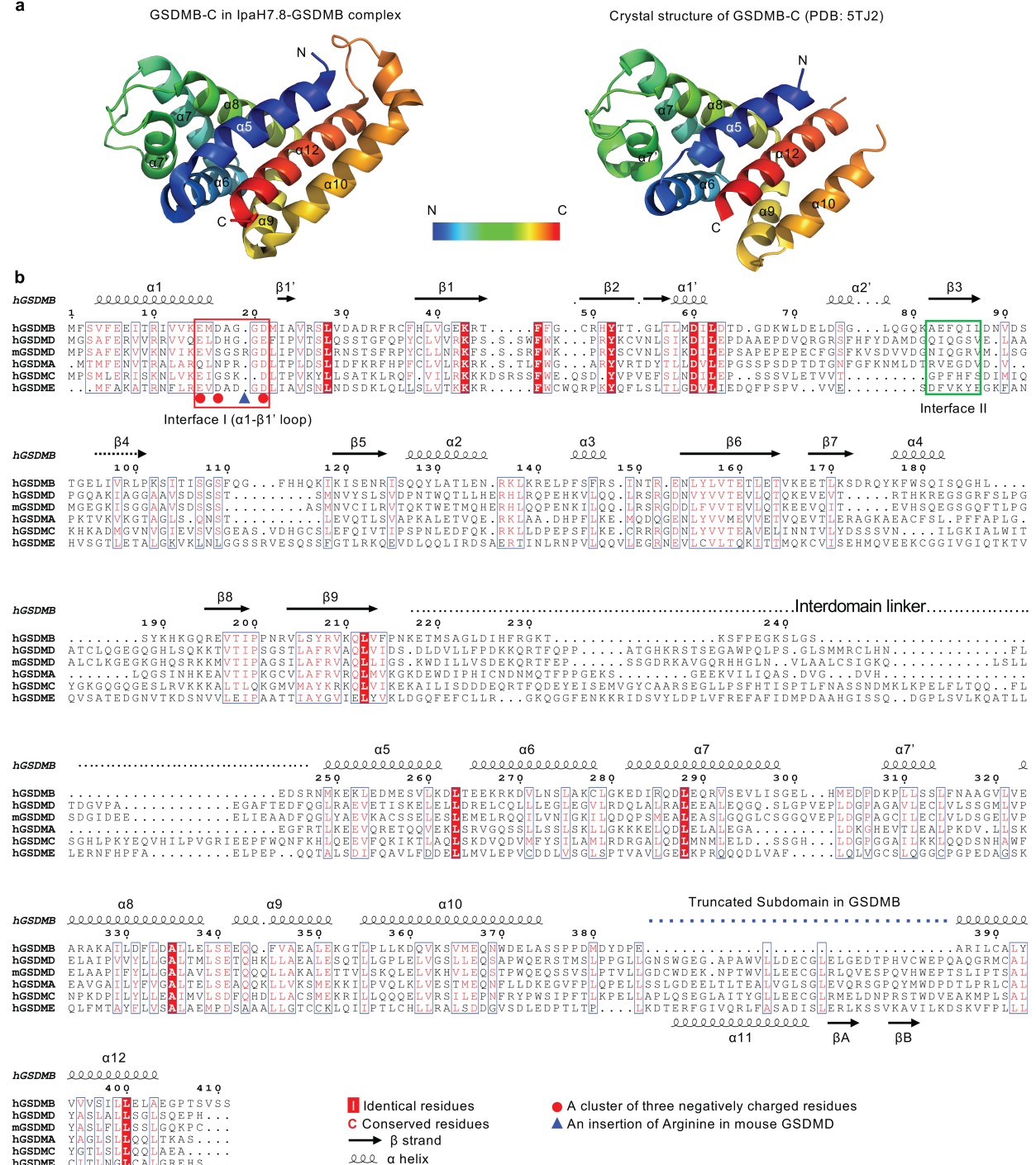

**Extended Data Fig. 2 | Cryo-EM structure of IpaH7.8-GSDMB complex.**
**a**, Structural comparison of GSDMB-C in the IpaH7.8-GSDMB complex and the reported crystal structure (PDB: 5TJ2). Colors are ramped from blue at the N terminus to red at the C terminus. **b**, Structure-based sequence alignment of human GSDMs and mouse GSDMD. Previously reported crystal structures of hGSDMD (PDB: 6N9O) and mGSDMD (PDB: 6N9N), and AlphaFold2 predicted structures of human GSDMA (hGSDMA), human GSDMC (hGSDMC), and human GSDME (hGSDME) are aligned against GSDMB using PROMALS3D. Secondary structural elements of GSDMB are indicated above the sequence. The interdomain linker and the C-terminal subdomain are also indicated. The two structural elements interacting with IpaH7.8 are marked by red and green boxes, respectively. The three negatively charged residues in the α1-β' loop are indicated by red dots, and the arginine insertion (mR20) in mouse GSDMD is indicated by a blue triangle.

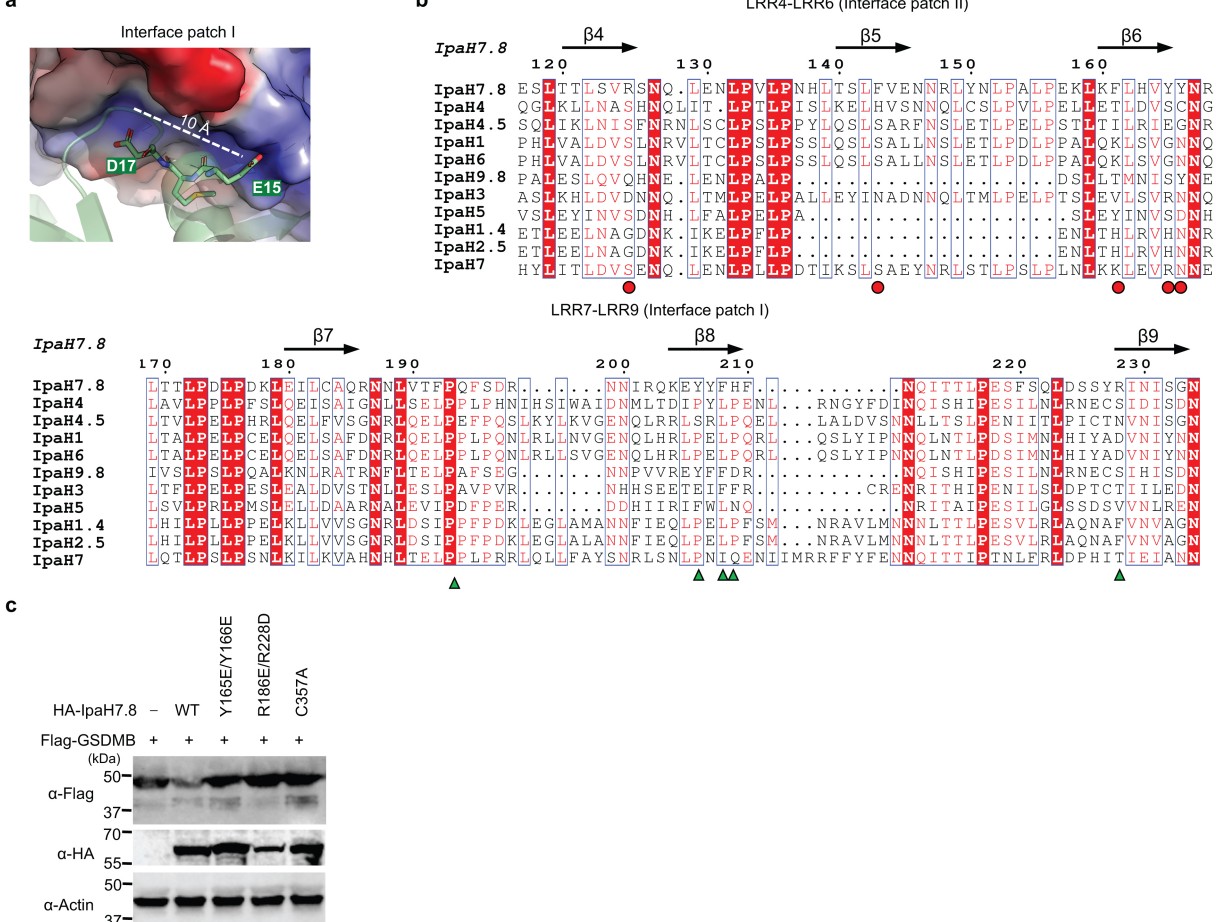

**Extended Data Fig. 3 | Interactions between GSDMB and IpaH7.8. a**, A close-up view of Interaction Patch I in the GSDMB-IpaH7.8 complex. GSDMB is shown as a ribbon diagram. Two negatively charged residues are shown as sticks. IpaH7.8 is shown as electrostatic potentials. The distance between the two small basic pockets formed by R186 and H209 and the surrounding residues in IpaH7.8 is labeled. **b**, Sequence alignment of the LRR domains of *Shigella* IpaHs. Secondary structural elements are indicated above the sequence. Universally conserved residues are marked with red shade, and partially conserved residues are colored red. Residues involved in Interface Patch II and I are indicated by red dots and green triangles, respectively. **c**, Immunoblots of 293T cells co-transfected with FLAG-tagged GSDMB and HA-IpaH7.8 (WT or indicated mutants). Results representative of 3 independent experiments.

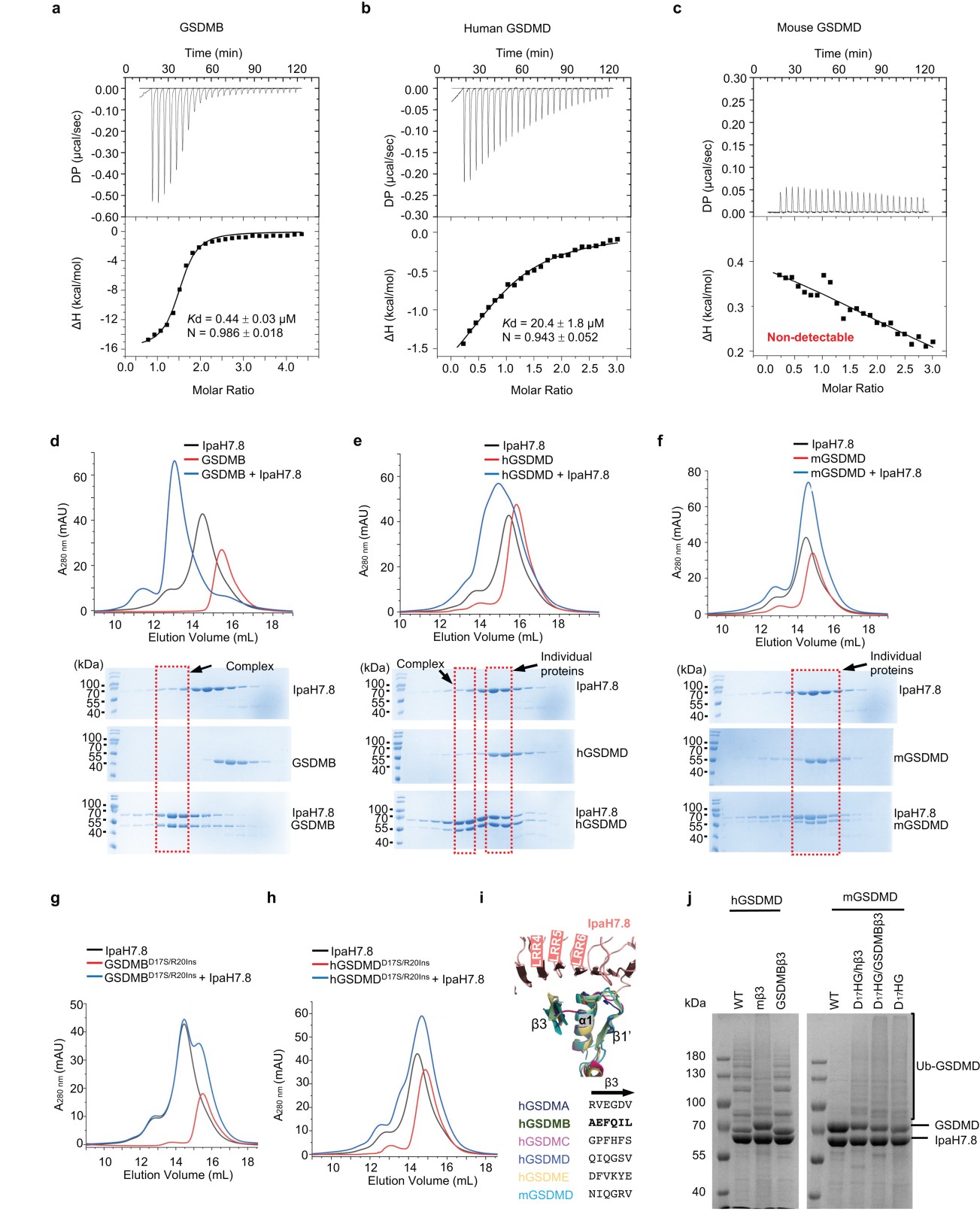

**Extended Data Fig. 4** | See next page for caption.

**Extended Data Fig. 4 | The three-negatively-charged-residue motif is the structural determinant in GSDM recognized by IpaH7.8. a–c**, ITC-based measurement of the binding affinities of IpaH7.8 with GSDMB (**a**), human (hGSDMD) (**b**), and mouse GSDMD (mGSDMD) (**c**), respectively. $K_d$, dissociation constant; N, the stoichiometry of the complex. DP, differential power measured by the ITC machine; ΔH, heat change measured by the ITC machine. The mean ± SD is shown (n = 3). **d–f**, Gel filtration profiles and Coomassie blue-stained SDS-PAGEs of IpaH7.8 incubated with GSDMB (**d**), human GSDMD (hGSDMD) (**e**), or mouse GSDMD (mGSDMD) (**f**). Results representative of more than 3 independent experiments. **g,h**, Gel filtration profiles IpaH7.8 incubated with GSDMB- (**g**) or human GSDMD-D17S/R20Ins (**h**) mutants, respectively. Results representative of 3 independent experiments. **i**, A close-up view of the interface between IpaH7.8 and GSDMs. Structures of human GSDMA (hGSDMA; AlphaFold2 predicted), human GSDMC (hGSDMC; AlphaFold2 predicted), human GSDMD (PDB: 6N9O), human GSDME (hGSDME; AlphaFold2 predicted), and mouse GSDMD (PDB: 6N9N) are superposed onto GSDMB in the structure of GSDMB-IpaH7.8 complex. The amino acid sequences of the β3 strand interacting with IpaH7.8 from each GSDM are shown. **j**, Coomassie blue-stained SDS-PAGE of *in vitro* ubiquitination of human and mouse GSDMD (WT and indicated mutants). mβ3, replaces the β3 strand in human GSDMD with the corresponding mouse sequence. GSDMBβ3, replaces the β3 strand in mouse GSDMD with the corresponding GSDMB sequence. Mutations did not alter the ubiquitination *in vitro*. Results representative of 3 independent experiments.

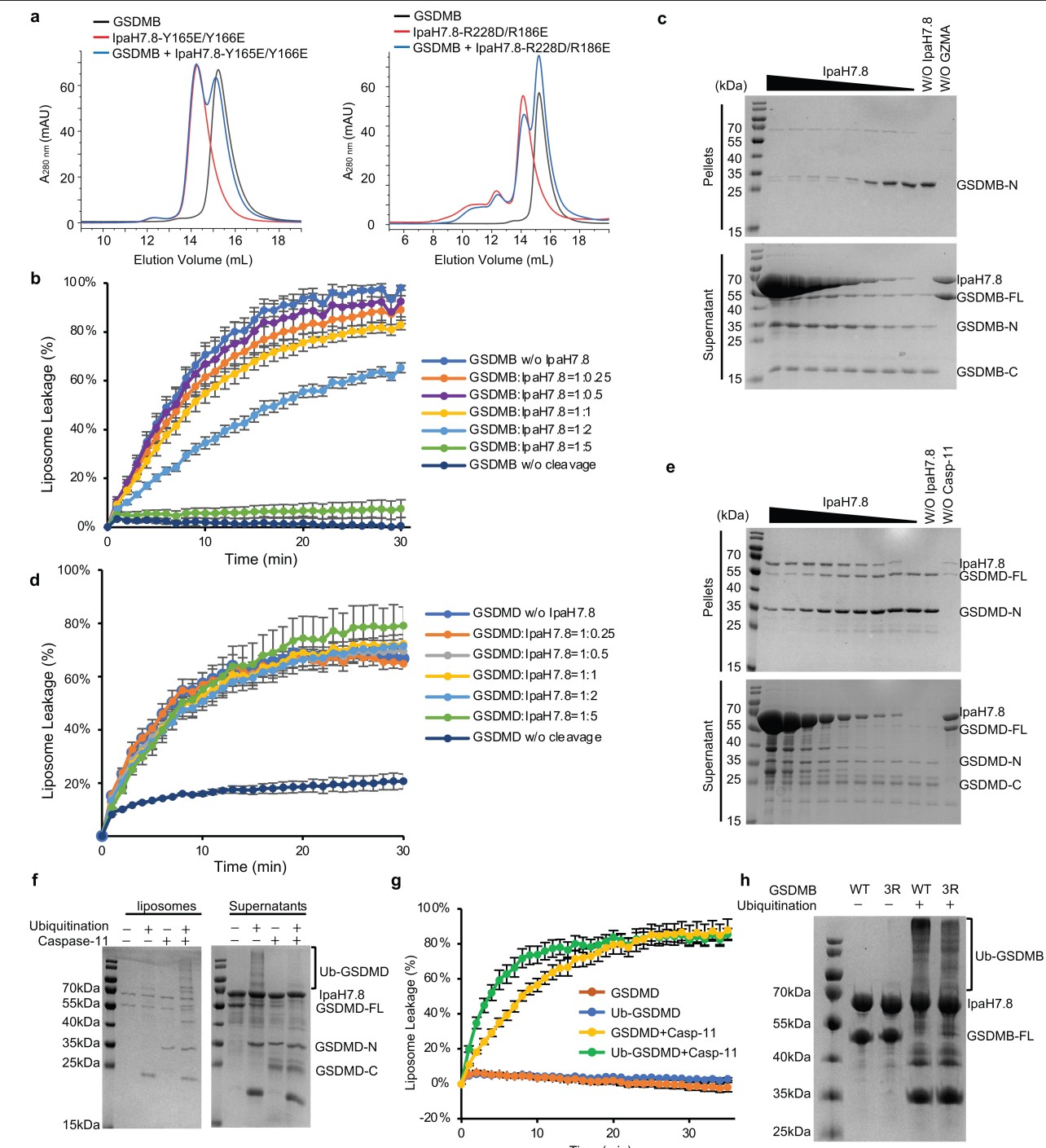

**Extended Data Fig. 5 | IpaH7.8 inhibits GSDMB pore formation. a**, Gel filtration profiles indicate no interaction between GSDMB and IpaH7.8-Y165E/Y166E mutant, or IpaH7.8-R186E/R228D mutant. Results representative of more than 3 independent experiments. **b**, The ability of GSDMB to induce liposome leakage when incubated with IpaH7.8 (WT) at different doses. Each dot represents the mean ± SD of 3 technical replicates. **c**, GSDMB association with CL-liposomes in the presence of IpaH7.8 in a liposome sedimentation assay. FL: full-length; N: N-terminal domain; and C: C-terminal domain. SDS-PAGEs were stained with Coomassie blue. Results representative of 3 independent experiments. **d**, The ability of human GSDMD to induce liposome leakage of CL-liposomes in the presence of IpaH7.8 at different doses. Each dot represents the mean ± SD of 3 technical replicates. **e**, Liposome sedimentation assay showing the association of human GSDMD with CL-liposomes in the presence of IpaH7.8. Results representative of 3 independent experiments. **f**, Liposome sedimentation assays show the association of ubiquitinated- or non-ubiquitinated human GSDMD with CL-liposomes. Results representative of 3 independent experiments. **g**, Liposome leakage assays show the effect of ubiquitination in inhibiting pore-forming activities of human GSDMD. The data were normalized with the fluorescence observed after adding detergent, and setting at zero of the fluorescence right before protein addition. Each dot represents the mean ± SD of 3 technical replicates. **h**, *In vitro* ubiquitination of GSDMB mutants with lysines mutated into arginines. 3R, with K177, K192, and K192 mutated into arginines in GSDMB. SDS-PAGEs were stained with Coomassie blue. Results representative of 3 independent experiments.

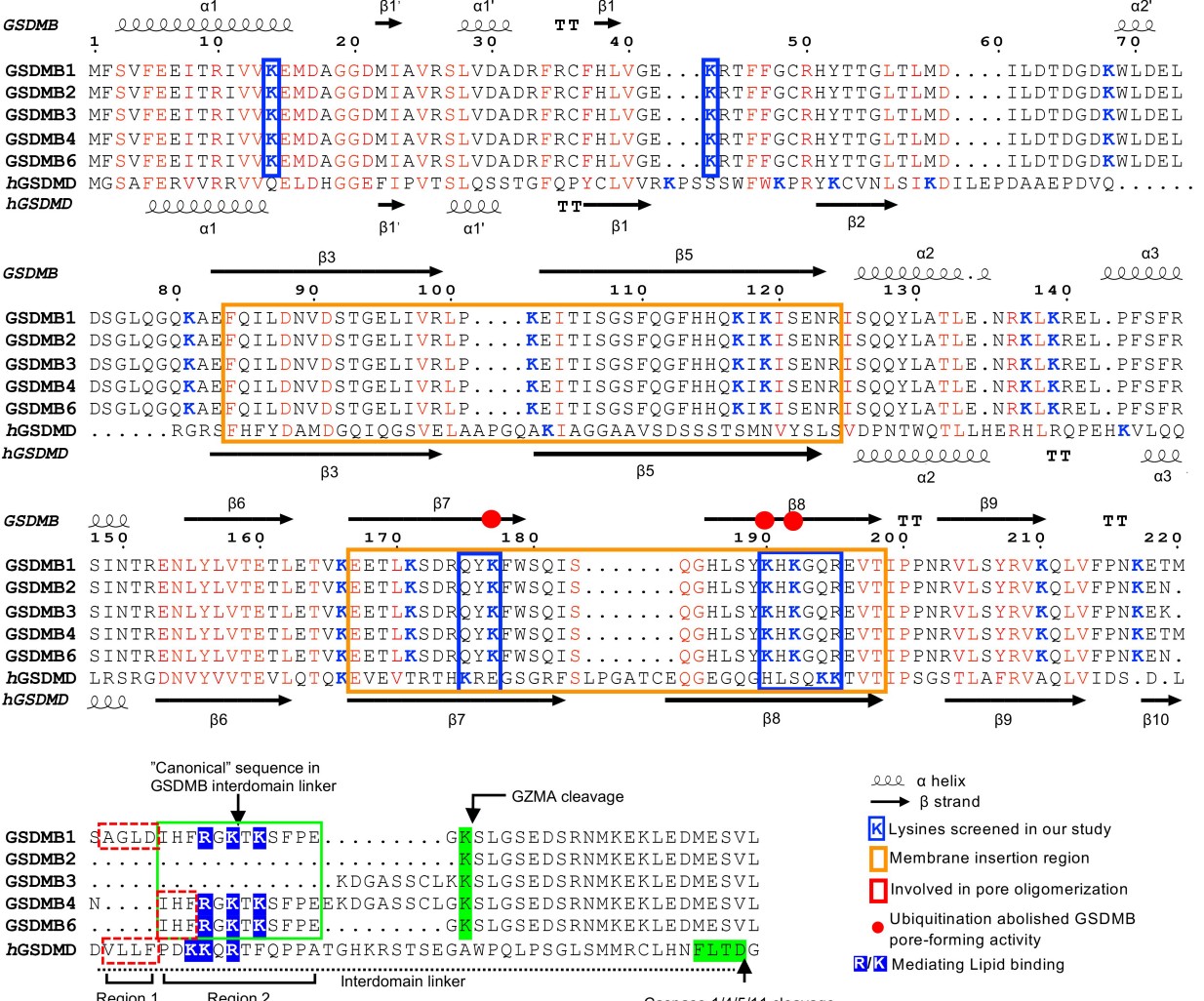

**Extended Data Fig. 6 | Structure-based sequence alignment of GSDMB and human GSDMD in their pore confirmation.** Secondary structural elements of GSDMB and human GSDMD (PDB: 6VFE) are indicated above and below the sequence, respectively. The two transmembrane hairpins are highlighted in orange boxes. All lysines in both GSDMB and human GSDMD are colored blue. Lysines predicted to be involved in pore-formation and screened for ubiquitination in our study are highlighted in blue boxes. The "canonical" sequence in the GSDMB interdomain linker is highlighted in a green box. The residues in the interdomain linker that may mediate pore oligomerization and lipid binding are highlighted in red dashed-line boxes and with blue background, respectively.

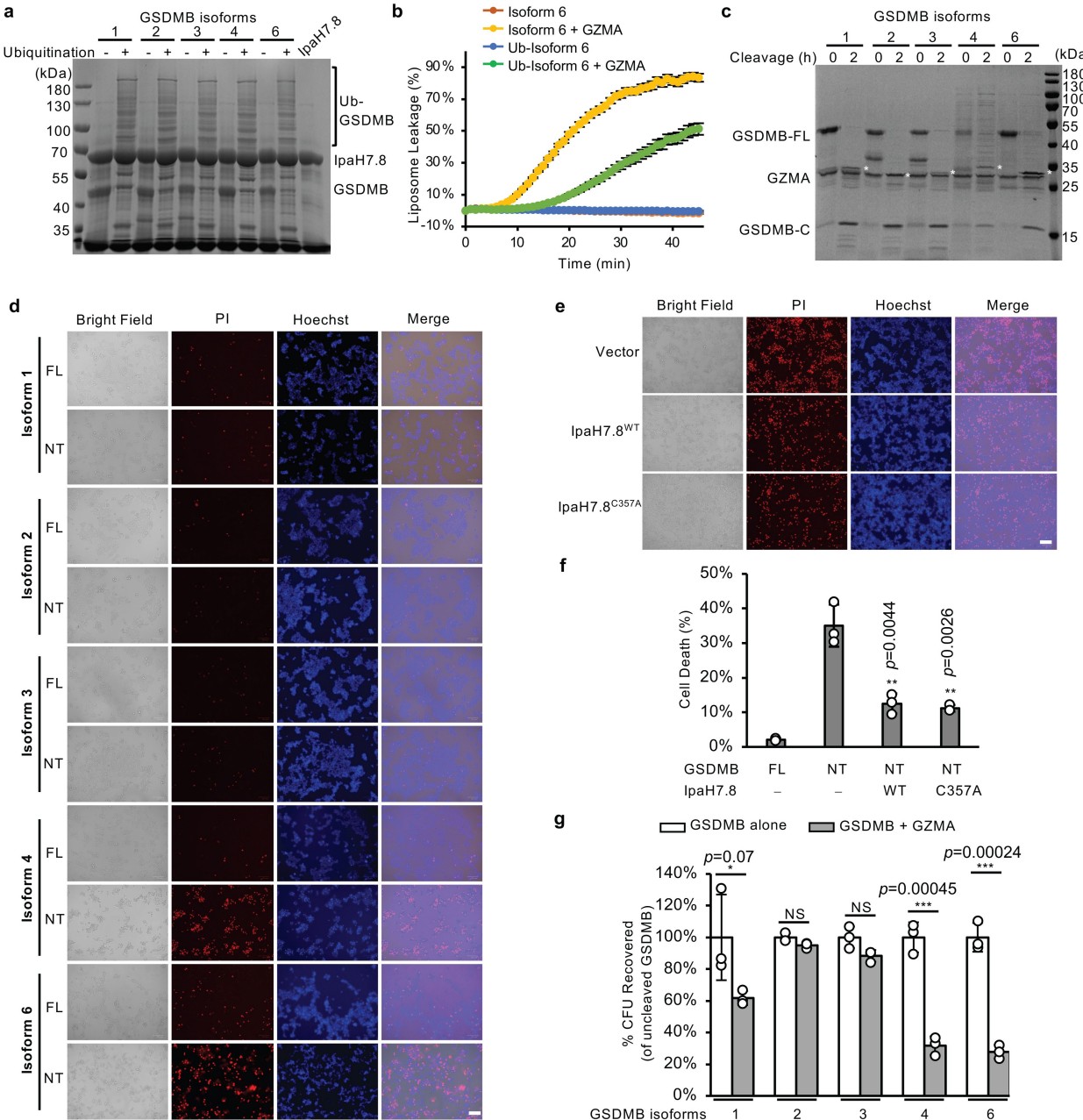

**Extended Data Fig. 7 | GSDMB isoforms are targeted equally by *Shigella* IpaH7.8 and GZMA. a**, *In vitro* ubiquitination of GSDMB isoforms by *Shigella* IpaH7.8. Ubiquitination reactions terminated at 0 min by boiling with SDS-PAGE loading buffer were used as non-ubiquitinated negative controls. SDS-PAGE was stained with Coomassie blue. Results representative of 3 independent experiments. **b**, Effect of ubiquitination in inhibiting pore-forming activities of GSDMB isoform 6 showed by liposome leakage assay using CL-liposomes. Each dot represents the mean ± SD of 3 technical replicates. **c**, Cleavage of GSDMB isoforms by GZMA. 3.6 µg of GSDMB isoforms were incubated with 1 µg of GZMA, respectively. Cleavage was carried out by incubating the mixture at 37 °C for 2 hours. White * indicates the GSDMB-NT. The molecular weight of GSDMB-NT of isoform 2 (25.9 kDa) is very close to GZMA (25.8 kDa) and cannot be separated on SDS-PAGE. SDS-PAGE was stained with Coomassie blue. Results

representative of 3 independent experiments. **d**, Effect of GSDMB isoforms in inducing pyroptosis in HEK293T cells. FL: full length GSDMB; NT: N-terminal domain of GSDMB. Scale bar is 100 µm. Results representative of 3 independent experiments. **e**, Effect of IpaH7.8 in inhibiting GSDMB isoform 4-mediated pyroptotic cell death of HEK293T cells. Results representative of 3 independent experiments. Scale bar is 100 µm. **f**, Quantification of cell death in (**e**). Error bars, mean ± SD of 3 independent experiments. One-way ANOVA followed by Tukey's post-hoc test compared to control of HEK293T cells transfected with plasmids of GSDMB-NT and empty vector. ** p < 0.005. **g**, Effect of GSDMB isoforms in inhibiting bacterial growth. Error bars, mean ± SD of 3 independent experiments. One-way ANOVA followed by Tukey's post-hoc test with samples compared to their uncleaved controls. *, p < 0.1, ***, p < 0.001, NS: not significant.

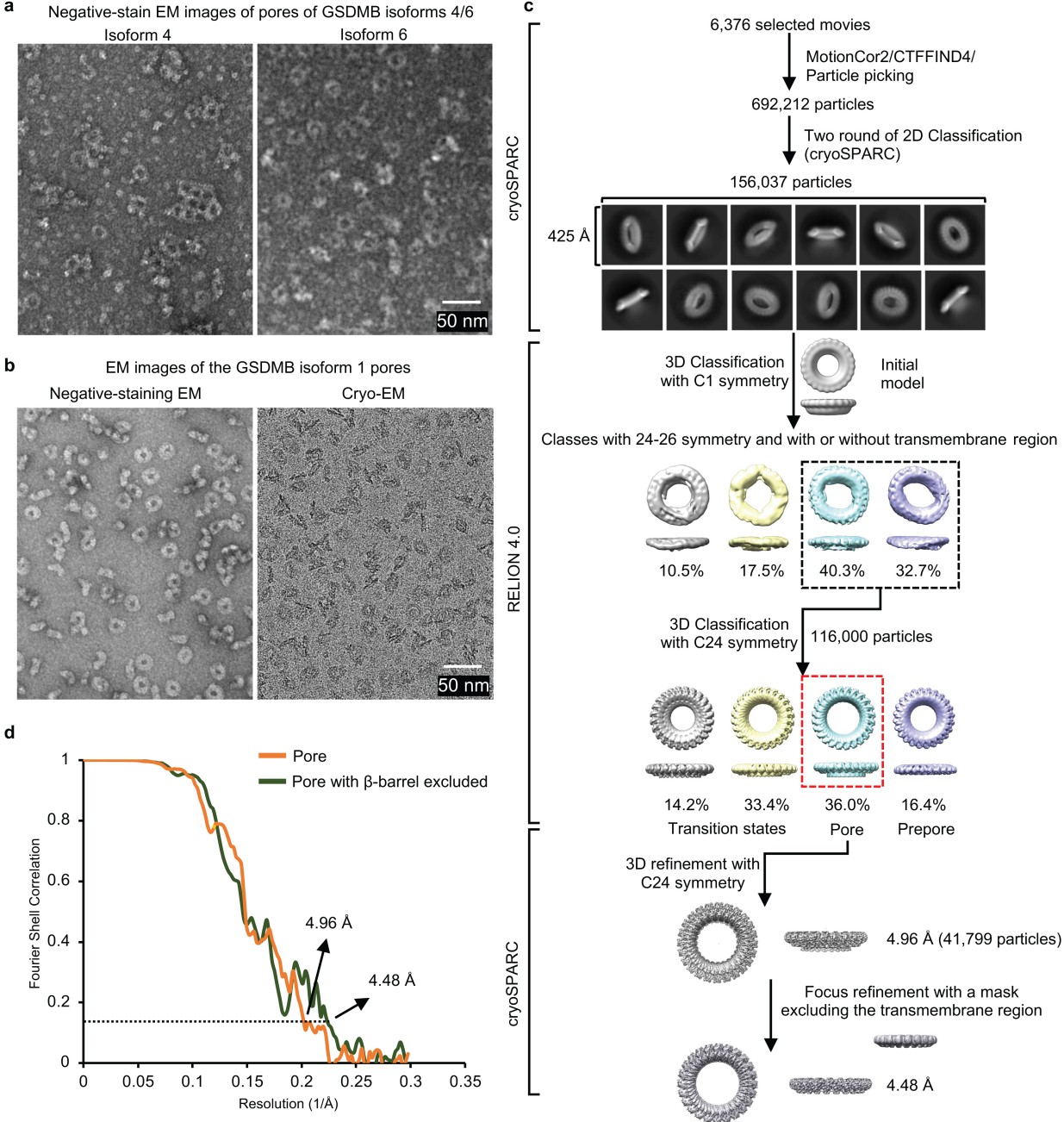

**a** Negative-stain EM images of pores of GSDMB isoforms 4/6

Isoform 4 · Isoform 6

cryoSPARC

50 nm

**b** EM images of the GSDMB isoform 1 pores

Negative-staining EM · Cryo-EM

50 nm

**c**

6,376 selected movies

MotionCor2/CTFFIND4/ Particle picking

692,212 particles

Two round of 2D Classification (cryoSPARC)

156,037 particles

425 Å

3D Classification with C1 symmetry · Initial model

Classes with 24-26 symmetry and with or without transmembrane region

10.5% · 17.5% · 40.3% · 32.7%

3D Classification with C24 symmetry · 116,000 particles

14.2% · 33.4% · 36.0% · 16.4%
Transition states · Pore · Prepore

3D refinement with C24 symmetry

4.96 Å (41,799 particles)

Focus refinement with a mask excluding the transmembrane region

4.48 Å

RELION 4.0

cryoSPARC

**d**

Fourier Shell Correlation vs Resolution (1/Å)

Pore
Pore with β-barrel excluded

4.96 Å
4.48 Å

**Extended Data Fig. 8 | Cryo-EM structural determination of GSDMB pore. a**, Representative negative stain-EM images of pores of GSDMB isoforms 4 and 6 extracted from cardiolipin-liposomes using detergent C12E8. Scale bar: 50 nm. Results representative of more than 3 independent experiments. **b**, A representative negative stain-EM image GSDMB isoform 1 pores solubilized in C12E8 (left panel) and a cryo-EM image of GSDMB isoform 1 pores collected on a Titan Krios microscope equipped with a K3 camera. Results representative of more than 3 independent experiments. Scale bar: 50 nm. **c**, A brief flow chart of single-particle cryo-EM data collection and process of GSDMB pore dataset. **d**, The gold-standard Fourier Shell Correlation curve for the half-map correlations of the GSDMB pore.

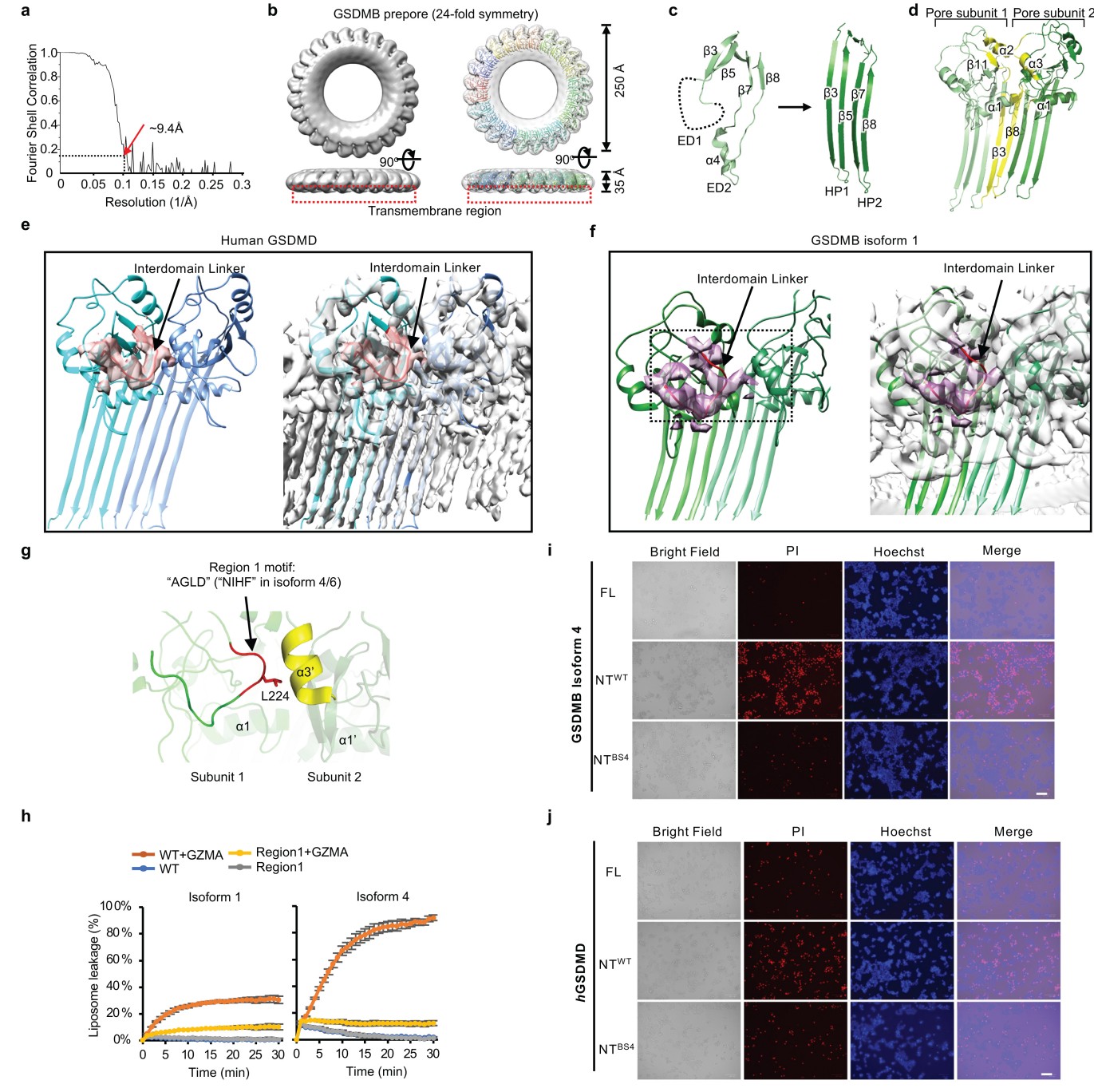

**Extended Data Fig. 9 | Structural basis for GSDMB pore formation. a**, The gold-standard FSC curve for the GSDMB prepore. **b**, The cryo-EM map (left) and the cryo-EM map fitted with an atomic model (right) of 24-fold GSDMB prepore. GSDMB protomers are colored differently. GSDMB prepore doesn't have the transmembrane insertion region. Red-dashed boxes indicated the missing transmembrane region. **c**, GSDMB pore β-hairpin (HPs) formation. The β3-β5 region from the first extension domain (ED1) transforms into HP1. The β7-α4-β8 region represents ED2 and turns into HP2. **d**, Two neighboring subunits in the GSDMB pore. Structural elements that participate in oligomerization are labeled and colored yellow. **e,f**, Cryo-EM densities of the interdomain linker in hGSDMD pore (**e**) and GSDMB pore (**f**) are shown, respectively. Overall densities are colored grey with the densities of the interdomain linkers in hGSDMD and GSDMB pores colored pink and magenta, respectively.

The interdomain linkers are colored red. **g**, A close-up view of the interdomain linker Region 1 in GSDMB isoform 1. Helix α3' form the neighboring subunit interacting with the interdomain linker is colored yellow. **h**, Effect of mutation of Region 1 in the interdomain linker of GSDMB isoforms 1 and 4 in inducing liposome (10% PS) leakage. Each dot represents the mean ± SD of 3 technical replicates. WT: wild type GSDMB; Region 1: mutation of Region 1 motif to "GGGG" in GSDMB. **i,j**, HEK293T cells were transiently transfected with indicated constructs of GSDMB isoform 4 (**i**) and hGSDMD (**j**) for 24 h and stained with Hoechst 33342 and PI. FL: full length; NT^WT: wild type GSDMB/ D-NT; NT^BS4: GSDMB/D-NT harboring a triple mutation of the three basic residues in GSDMB isoform 4 and in hGSDMD to glutamic acids in BS4. Results representative of 3 independent experiments. Scale bar is 100 μm.

## Extended Data Table 1 | Cryo-EM data collection, refinement and validation statistics

| | GSDMB-IpaH7.8 (EMDB: EMD-28087) (PDB: 8EFP) | GSDMB pore (EMDB: EMD-28584) (PDB: 8ET2) | GSDMB pore [w/o β-barrel] (EMDB: EMD-28583) (PDB: 8ET1) |
|---|---|---|---|
| **Data collection and processing** | | | |
| Magnification | 105,000 | 105,000 | |
| Voltage (kV) | 300 | 300 | |
| Electron exposure (e$^-$/Å$^2$) | 68.44 | 50 | |
| Defocus range (μm) | -0.8~-2.5 | -1.0~-2.0 | |
| Pixel size (Å) | 0.828 | 0.83 | |
| Initial particles (No.) | 1,522,742 | 692,212 | |
| Final particles (No.) | 113,959 | 41,799 | |
| Map Resolution (Å) | 3.8 | 4.96 | 4.48 |
| FSC threshold | 0.143 | 0.143 | 0.143 |
| Map resolution range (Å) | 30~3.8 | 30~4.96 | 30~4.48 |
| **Refinement** | | | |
| Model resolution (Å) | 3.9 | 5.6 | 4.9 |
| FSC threshold | 0.143 | 0.143 | 0.143 |
| Map CC | 0.83 | 0.69 | 0.79 |
| Map sharpening B factor (Å$^2$) | 150.2 | 313.4 | 200.1 |
| Model composition | | | |
| Non-hydrogen atoms | 4764 | 43584 | 35976 |
| Protein residues | 594 | 5304 | 4368 |
| R.m.s. deviations | | | |
| Bond lengths (Å) | 0.006 | 0.004 | 0.003 |
| Bond angles (°) | 1.194 | 1.001 | 0.817 |
| Validation | | | |
| MolProbity score | 2.66 | 2.46 | 2.55 |
| Clashscore | 33.4 | 23.8 | 29.82 |
| Poor rotamers | 0 | 0 | 0 |
| Ramachandran plot | | | |
| Favored (%) | 86 | 88.02 | 88.51 |
| Allowed (%) | 14 | 11.98 | 11.49 |
| Outlier (%) | 0 | 0 | 0 |

Cryo-EM map of GSDMB pore[w/o β-barrel] were obtained from the map of GSDMB pore through the focus refinement with a mask excluding the transmembrane β-barrel region.

# Reporting Summary

## Statistics

For all statistical analyses, confirm that the following items are present in the figure legend, table legend, main text, or Methods section.

| n/a | Confirmed | |
|---|---|---|
| ☐ | ☒ | The exact sample size (*n*) for each experimental group/condition, given as a discrete number and unit of measurement |
| ☐ | ☒ | A statement on whether measurements were taken from distinct samples or whether the same sample was measured repeatedly |
| ☐ | ☒ | The statistical test(s) used AND whether they are one- or two-sided *Only common tests should be described solely by name; describe more complex techniques in the Methods section.* |
| ☒ | ☐ | A description of all covariates tested |
| ☒ | ☐ | A description of any assumptions or corrections, such as tests of normality and adjustment for multiple comparisons |
| ☐ | ☒ | A full description of the statistical parameters including central tendency (e.g. means) or other basic estimates (e.g. regression coefficient) AND variation (e.g. standard deviation) or associated estimates of uncertainty (e.g. confidence intervals) |
| ☒ | ☐ | For null hypothesis testing, the test statistic (e.g. *F*, *t*, *r*) with confidence intervals, effect sizes, degrees of freedom and *P* value noted *Give P values as exact values whenever suitable.* |
| ☒ | ☐ | For Bayesian analysis, information on the choice of priors and Markov chain Monte Carlo settings |
| ☒ | ☐ | For hierarchical and complex designs, identification of the appropriate level for tests and full reporting of outcomes |
| ☒ | ☐ | Estimates of effect sizes (e.g. Cohen's *d*, Pearson's *r*), indicating how they were calculated |

*Our web collection on statistics for biologists contains articles on many of the points above.*

## Software and code

Policy information about availability of computer code

| Data collection | We used SerialEM for cryo-EM data collection. |
|---|---|
| Data analysis | We used MotionCor2, CTFFIND4, Relion 4.0, cryoSPARC v3.3.1, crYOLO 1.7.6, Coot 0.9.8.4, PHENIX 1.20-4487, PyMol 2.5.3, ResMap 1.95, Molprobity 4.02-528 and UCSF Chimera 1.15 for cryo-EM data analysis, and used Origin 7.0 for ITC data analysis |

For manuscripts utilizing custom algorithms or software that are central to the research but not yet described in published literature, software must be made available to editors and reviewers. We strongly encourage code deposition in a community repository (e.g. GitHub). See the Nature Portfolio guidelines for submitting code & software for further information.

## Data

Policy information about availability of data

All manuscripts must include a data availability statement. This statement should provide the following information, where applicable:
- Accession codes, unique identifiers, or web links for publicly available datasets
- A description of any restrictions on data availability
- For clinical datasets or third party data, please ensure that the statement adheres to our policy

The atomic coordinates of the GSDMB-IpaH7.8 complex, GSDMB pore, GSDMB pore without β-barrel have been deposited in the Protein Data Bank (PDB) under accession numbers 8EFP, 8ET2, and 8ET1, respectively. The associated cryo-EM density maps have been deposited in the Electron Microscopy Data Bank (EMDB) under accession numbers EMD-28087, EMD-28584, and EMD-28583, respectively. All other data, for example, the atomic coordinate of GSDMB prepore which is

not deposited because of the low resolution, are available from the corresponding author upon request. Several structural coordinates in the PDB database were used in this study, which can be located by accession numbers 6CB8, 5B5R, 6N9O, 6N9N, 6VFE, 7V8H, and 3CVR.

## Human research participants

Policy information about studies involving human research participants and Sex and Gender in Research.

| | |
|---|---|
| Reporting on sex and gender | N/A |
| Population characteristics | N/A |
| Recruitment | N/A |
| Ethics oversight | N/A |

Note that full information on the approval of the study protocol must also be provided in the manuscript.

# Field-specific reporting

Please select the one below that is the best fit for your research. If you are not sure, read the appropriate sections before making your selection.

☒ Life sciences      ☐ Behavioural & social sciences      ☐ Ecological, evolutionary & environmental sciences

For a reference copy of the document with all sections, see nature.com/documents/nr-reporting-summary-flat.pdf

# Life sciences study design

All studies must disclose on these points even when the disclosure is negative.

| | |
|---|---|
| Sample size | Sample sizes were not pre-determined. Cryo-EM images were collected until structures of satisfactory quality were solved, which suggested sufficient sample size. For biochemical and cellular experiments, no information was derived about a population based on sampling, and therefore sample size determination was not necessary. |
| Data exclusions | In cryo-EM processing, we discarded "junk" particles that could not be classified into useful 3D reconstructions. This is a widely used and accepted practice in the cryo-EM field. No other data were excluded from analysis. |
| Replication | All experiments were performed independently at three times with similar results, as described in the figure legends. |
| Randomization | Proteins, liposomes, and cells were randomly allocated to the wells in each experimental group. Other randomization of experimental groups was not relevant to this study, and independent variables were controlled and did not require randomization. |
| Blinding | Blinding was not performed as subjective analysis was not needed. Each experiment was analyzed using consistent methods. Random allocation and quantitative measurements using various approaches and reaction kits as described in the methods minimized biased assessments. |

# Reporting for specific materials, systems and methods

We require information from authors about some types of materials, experimental systems and methods used in many studies. Here, indicate whether each material, system or method listed is relevant to your study. If you are not sure if a list item applies to your research, read the appropriate section before selecting a response.

## Materials & experimental systems

| n/a | Involved in the study |
|---|---|
| ☐ | ☒ Antibodies |
| ☐ | ☒ Eukaryotic cell lines |
| ☒ | ☐ Palaeontology and archaeology |
| ☒ | ☐ Animals and other organisms |
| ☒ | ☐ Clinical data |
| ☒ | ☐ Dual use research of concern |

## Methods

| n/a | Involved in the study |
|---|---|
| ☒ | ☐ ChIP-seq |
| ☒ | ☐ Flow cytometry |
| ☒ | ☐ MRI-based neuroimaging |

## Antibodies

| Antibodies used | Anti-ubiquitin (Thermo Fisher Scientific, PA3-16717, Lot:XG344606, 1:1000)<br>Anti-FLAG (Sigma-Aldrich, F1804, Source#: SLCM4081, 1:1000)<br>Anti-HA (Cell Signaling Technology, 3724S, Lot: 9, 1:1000)<br>Anti-actin (Cell Signaling Technology, 3700S, Lot: 20, 1:1000)<br>Peroxidase AffiniPure Goat Anti-Mouse IgG (H+L) (Jackson Immuno Research Inc., 115-035-166, Lot: 155426, 1:5000)<br>Peroxidase-AffiniPure Goat Anti-Rabbit IgG (H+L) (Jackson Immuno Research Inc., 115-035-144, Lot: 163357, 1:5000) |
|---|---|
| Validation | All antibodies used in this study are commercially available and have been validated by the manufacturers' and/or previous publications.<br>Anti-ubiquitin (https://www.thermofisher.com/antibody/product/Ubiquitin-Antibody-Polyclonal/PA3-16717. PMID: 30547882)<br>Anti-FLAG (https://www.sigmaaldrich.com/US/en/product/sigma/a8592. PMID: 26727110, 25744187, 20980514, 20356955, 19153083, 18403418, etc)<br>Anti-HA (https://www.cellsignal.com/products/primary-antibodies/ha-tag-c29f4-rabbit-mab/3724. PMID: 27043414, 36307403, 36127332, 35918345, 35908039, 35550517, 34819506, 33972784, etc)<br>Anti-actin (https://www.cellsignal.com/products/primary-antibodies/b-actin-8h10d10-mouse-mab/3700. PMID: 36216837, 35831316, 35672408, 35610475, 35588457, 35602949, 35143048, 35487895, 35332119, etc)<br>Peroxidase AffiniPure Goat Anti-Mouse IgG (H+L)  (https://www.jacksonimmuno.com/catalog/products/115-035-166. PMID: 36543799, 35505004, 35658004, 36072551, etc)<br>Peroxidase-AffiniPure Goat Anti-Rabbit IgG (H+L) (https://www.jacksonimmuno.com/catalog/products/111-035-144. PMID: 36109647, 36563856, 36543799, etc) |

## Eukaryotic cell lines

Policy information about cell lines and Sex and Gender in Research

| Cell line source(s) | HEK293T cells were obtained from American Type Culture Collection (ATCC). |
|---|---|
| Authentication | HEK293T cells were authenticated by ATCC. |
| Mycoplasma contamination | All cell lines were tested to be mycoplasma-negative by PCR. |
| Commonly misidentified lines<br>(See ICLAC register) | No commonly misidentified cell lines were used in this study. |

