## [Peer Review File · Nature]

Manuscript Title: Structural basis for GSDMB pore formation and its targeting by a bacterial effector

Reviewer Comments & Author Rebuttals

Reviewer Reports on the Initial Version:

Referees' comments:

Referee #1 (Remarks to the Author):

Structural basis for a unique regulation of GSDMB pore formation and the recognition of GSDMB by Shigella effector IpaH7.8

Wang et al.

Summary:

The present study determines the structural basis of Gasdermin B (GSDMB) targeting by Shigella-encoded IpaH7.8, and how alternative splicing of GSDMB regulates its pore-forming activity. To achieve this, the investigators solve Cryo-EM structures of full-length (FL) GSDMB in a complex with IpaH7.8, or the entire GSDMB pore. A combination of in vitro functional assays are used to support their structural studies.

First, by focusing on the GSDMB-IpaH7.8 complex, the investigators interrogate the mechanism of IpaH7.8 selectivity for GSDMB and GSDMD. From this analysis, an acidic motif of the GSDMB N-terminus (E15, D17, and D21) is found to interact extensively with the IpaH7.8 N-terminal Leucine-Rich Repeat (LRR) domain. A homologous motif is conserved in human GSDMD (susceptible to IpaH7.8), but absent in mouse GSDMD (resistant to IpaH7.8). These results confirm a previous study that mapped the minimal susceptibility region of GSDMD to include the same acidic patch (Luchetti et al.). In contrast to the previous study that showed IpaH7.8 could bind both human and mouse GSDMD, the current report could not detect binding for the mouse protein. In contrast to GSDMD, IpaH7.8 blocked GSDMB pore-forming activity by two mechanisms, binding and ubiquitination. Ubiquitination of GSDMB lysine residues absent in GSDMD disrupted key binding interfaces of the GSDMB beta barrel within the pore.

To reconcile why previous reports on GSDMB differ in their conclusions about its pore-forming activity, the investigators characterize each GSDMB isoform (iso-1-6) separately. Of these, GSDMB iso-5 excludes the essential N-terminal pore-forming domain. For the remaining isoforms, each differ primarily by inclusion or exclusion of GSDMB Exon 6 coding residues (N221-E233) of the inter-domain linker. Of these, only GSDMB iso-4 and iso-6 contain the Exon 6 coding region, and each are found to exhibit potent pore-forming activity in vitro. In contrast, GSDMB iso-2 and iso-3 that lack the Exon 6 sequence are completely inactive. Lastly, as a result of non-canonical splicing, GSDMB iso-1 introduces 4 additional residues together with a single amino acid substitution (216SAGL220 and

N221D) upstream of the Exon 6 sequence. This GSDMB isoform is the one previously characterized to exclude its pyroptosis functions (Hansen et al. and Rana et al.). Interestingly, GSDMB iso-1 is shown to possess 20-30% weaker pore-forming activity than GSDMB iso-4/6.

Next, by solving the structure of the GSDMB iso-1 pore, the investigators interrogate the structural basis for its pore-forming activity. Despite GSDMB iso-1 being the weakest pore-forming isoform, the investigators successfully solve its cryo-EM structure. In this structure, the entire sequence of the inter-domain linker was resolved (V229-Q241). By comparison, the GSDMD pore structure contains a basic patch of residues (K235, K236, and R238) that are oriented toward the membrane bi-layer. The corresponding sequence in GSDMB iso-1 reveals how alternative splicing disrupts this lipid binding interface. Simultaneously, insertion of 216SAGLD221 shifts the basic patch encoded by Exon 6 away from the membrane bi-layer and provides repulsive interactions with proximal negatively charged lipids. Lastly, to confirm the importance of this basic patch in pore-forming competent GSDMB iso-4, the investigators showed how mutation of these residues attenuates its pore-forming activity.

Opinion:

The current study provides strong structural support for the underlying mechanisms that govern sensitivity of GSDMB/D to IpaH7.8 and how GSDMB alternative splicing regulates its pore-forming activity. However, in the latter case, the investigators rely solely on in vitro liposome leakage assays to functionally test their structural models. Since the controversy surrounding GSDMB pertains to its ability to lyse mammalian cells, this should not be excluded from their analysis. Despite this, the data presented do provide a compelling explanation for the observed patterns of GSDMB activity appearing in the literature.

While these findings are of general importance to the field of Gasdermin family regulation, many of its conclusions confirm what has already been reported by earlier studies. For example, GSDMB and GSDMD were already proven substrates of IpaH7.8 (Hansen et al. and Luchetti et al.). For GSDMD, the underlying sequence determinants were already mapped albeit without structural resolution. Extending these to GSDMB in the context of the structure, while interesting, adds little value to the field of bacterial pathogenesis.

As it pertains to the structures of full-length GSDMB and the GSDMB pore, aside from the specific structural roles played by alternative splicing, these structures confirm many of the regulatory mechanisms already reported for available structures of GSDMD and GSDMA3. For example, like GSDMD/A3, GSDMB overlaps significantly in the structural elements that determine its auto-inhibitory state and pore-assembly.

An important unanswered question is whether cells exert control over the patterns of GSDMB isoform expression to achieve specific physiological functions. For example, in the pioneering report by Zhou et al., IFN- γ was found to induce GSDMB expression. In light of the isoform-specific activities of distinct GSDMB isoforms, it would be interesting to examine how their relative levels of expression change in response to IFN- γ stimulation. By extension, are there unidentified pathways that differentially regulate distinct GSDMB isoforms? Finally, do cancer cells differentially regulate GSDMB isoforms as a resistance mechanism against NK/T-cell tumor immunity?

Integrated together, the current study is valuable to a more specialist audience. It does not present sufficient novelty for publication in Nature.

Specific Points:

1. Which GSDMB isoforms trigger pyroptosis in mammalian cells? i.e. LDH-release after cleavage of FL GSDMB by GZMA or upon expression of GSDMB-N fragments?
2. How do GSDMB-N isoforms affect bacterial viability when added exogenously? i.e. expression from within bacteria is non-physiological
3. Do Exon 6 coding sequences also mediate interactions between adjacent GSDMB monomers in the pore? If so, how do corresponding mutations impact GSDMB pore-forming activity?
4. How does mutation of GSDMD basic patch 4 impact its pore-forming activity?
5. Does the purified mouse GSDMD undergo cleavage by CASP4/11? If so, does this unleash its pore-forming activity?

Referee #2 (Remarks to the Author):

In this manuscript, Wang et al. reported the cryo-EM structures of both GSDMB N-terminal domain (NT) pore, and the complex of full-length GSDMB with the *Shigella* virulence effector IpaH7.8. The structure of the GSDMB-IpaH7.8 complex revealed that a three-negatively-charged-residue motif determines the recognition of GSDMB-NT by IpaH7.8, which explains the recognition of GSDMB and human GSDMD, and the lack of interaction with mouse GSDMD. Interestingly, the authors showed that the binding of IpaH7.8, as well as the ubiquitination of GSDMB (in residues within the transmembrane domain), directly inhibits GSDMB pore formation. This is by contrast to GSDMD whose ubiquitination by IpaH7.8 was shown previously not to be sufficient for inhibition of pore formation, but whose inhibition requires subsequent proteasomal degradation. The authors investigated the different GSDMB isoforms and showed that they exhibited different pore formation ability. The GSDMB pore structure they solved further revealed that the interdomain linker, which varies between isoforms, participates in lipid binding, which can explain the isoform-dependent differential activities. These studies are carefully performed and of high quality.

Specific comments:

Major:

- 1- Several in-vitro ubiquitination experiments such as Figure 2C, 2E, 2F, 2G, 2H, 2I and 2J should show the bottom part of the gel to demonstrate the unconjugated ubiquitin in the reaction. This is important for GSDMB mutations with decreased ubiquitination such as Y165A/Y166A or Y165E/Y166E, lanes 4 and 5 in Figure 2E. This information will prove that the ubiquitin was equally available in all reactions but was not conjugated in the case of the mutants. Ubiquitin is an 8.6 kDa protein and should be visible in a gradient gel SDS-PAGE.
- 2- Related to Figure 2F: The authors used a co-transfection system and observed loss of IpaH7.8-mediated degradation of certain GSDMB mutants. This observation should be supplemented with

ubiquitin blot showing ubiquitinated species on immunoprecipitated GSDMB as a second line of evidence that the proposed mutations alter IpaH7.8-mediated degradation of GSDMB.

3- Figure 2J lacks negative control for the poly-ubiquitination signal. To show specificity, catalytically inactive form of IpaH7.8 should be included in the analysis.

4- 3D-classification of cryo-EM datasets includes also the prepore structure of GSDMB. Models of the prepore structure should also be built and reported.

Minor

1- It should be mentioned in the legend of Figure 3G that the demonstrated image is a Coomassie blue-stained SDS-PAGE as it was mentioned in other figure legends.

2- The sequence arrangements in mouse GSDMD used in Figure 2H should be illustrated with a representative schema.

3- 3D-classification of GSDMB was referred in the text as Extended Figure Data 7C, however, this data is in the next figure (Extended Figure Data 8C). This should be corrected in the text.

Referee #3 (Remarks to the Author):

This paper discusses how a bacterial secretion system effector protein, Ipa7.8 from *Shigella*, binds to the human GSDMB protein. GSDMB is a member of the GSDM family of proteins, of which the most famous is GSDMD. These are proteins that are processed, and one cleavage fragment oligomerizes in the cell membrane to create a pore. GSDM pores are consistent with pyroptotic cell death. The biology of GSDMB is not clear, but it is believed that Granzyme B can cleave it for pore formation. *Shigella* Ipa7.8 is in the present report shown to interact with GSDMB in cryo-EM experiments. They also investigate the role of Ubiquitination of GSDMB. In addition, structures of the whole GSDMB pore is clarified, together with studies of various splicing variants of GSDMB. The effect of these aspects on liposome pore formation is shown. Taken together, these defined data sets represent a significant advance of the field.

Specific comments:

Overall the data are clear and it is very interesting to observe the direct binding of a bacterial inhibitor to a GSDM protein, this is a first. It is also a strength to have meaningful discussions about various GSDMB isoforms, this aspect of splicing is often neglected in many papers, although this paper could have covered more ground on various isoforms. How are these isoforms compare to what has been studied in previous papers?

Many of their observations are backed up with experiments with lysis of liposomes. However, I do miss a more direct assay for actual cell death. They express GSDMB in 293 cells for WB/IP type of experiments, but why are 293 cells not also used for cell death assays? In the right conditions, over-expression in 293 is a relatively quick and easy way to induce cell death. This type of assay, although still in an artificial system, would carry more weight and could lend further support to many of their conclusions on Ipa7.8 inhibition, isoforms etc, and this could be important.

Methodology seems valid. Appropriate statistical analyses are included. References are OK. Conclusions are well supported.

Fig 2 c: "single mutations of either E15 or D17 in GSDMB into a reversely 132 charged or uncharged residue completely or significantly abolished the ubiquitination of GSDMB 133 by IpaH7.8 in vitro" D17S seems more of a partial reduction, especially since Gsdmb band appears significantly reduced like WT. Perhaps the authors can be more careful with the language here.

Fig 3c : The bacterial growth inhibition assay was nice as another measure of GSDMB pore function. However it is quite artificial to place GSDMB into E coli. Exploring Gsdmb from the host (or artificial host) inserting into bacterial membranes would be great support.

The Ub-studies, in particular how they relate to species differences in GSDMB proteins, is a nice addition, especially the "rescue" of mouse GSDMB with human sequences.

Pore analysis: What is the reasoning behind the choice of isoform 1 for the Cryo-em pore analysis instead of 4 or 6 that had better permeabilization activity? Do they think other isoforms could give different pore compositions? Please address this question.

"We primarily focused on GSDMB isoform 1 because of its attenuated pore-forming activity"

But why is that better comparison to other GSDM proteins?

Please include a discussion on the potential importance of GSDMB isoforms on Shigella infection.

Author Rebuttals to Initial Comments:

We thank Reviewer #1 for the insightful comments and suggestions.

Opinion:

A) The current study provides strong structural support for the underlying mechanisms that govern sensitivity of GSDMB/D to IpaH7.8 and how GSDMB alternative splicing regulates its pore-forming activity. However, in the latter case, the investigators rely solely on *in vitro* liposome leakage assays to functionally test their structural models. Since the controversy surrounding GSDMB pertains to its ability to lyse mammalian cells, this should not be excluded from their analysis. Despite this, the data presented do provide a compelling explanation for the observed patterns of GSDMB activity appearing in the literature.

Response: We thank the reviewer for the comments and suggestions. We have now included cellular experiments to validate our structural models.

1. We have tested the cytotoxicity of GSDMB isoforms in HEK293 cells. Cell death was monitored by Hoechst/PI double staining. The results showed that isoforms 4 and 6 induced remarkable cell death, while isoforms 1, 2, and 3 did not. These data are included in the new Fig. 4b and Extended data Fig. 7d.
2. We also tested the IpaH7.8 inhibition in cells. As isoform 1 does not induced pyroptotic cell death, we co-transfected HEK293T cells with GSDMB isoform 4 and IpaH7.8 (WT or C357A mutant). The results showed that GSDMB isoform 4 mediated pyroptotic cell death was greatly inhibited by IpaH7.8. The inhibition is IpaH7.8 E3 ligase activity independent. These data confirm the direct inhibition of GSDMB by IpaH7.8 through binding. These data are included in the new Extended data Fig. 7e and f.
3. We also tested the effect of mutation of lipid binding site 4 (BS4) in HEK293T cells. Consistent with the *in vitro* liposome leakage assay, mutation significantly reduced the capability of GSDMB/D to induce cell death. These data are included in the new Fig. 5h and Extended data Fig. 10f and g.

While these findings are of general importance to the field of Gasdermin family regulation, many of its conclusions confirm what has already been reported by earlier studies. For example, GSDMB and GSDMD were already proven substrates of IpaH7.8 (Hansen et al. and Luchetti et al.). For GSDMD, the underlying sequence determinants were already mapped albeit without structural resolution. Extending these to GSDMB in the context of the structure, while interesting, adds little value to the field of bacterial pathogenesis.

Response: We thank the reviewer for the comments. Luchetti *et al.* did map a sequence in GSDM-N that is required for IpaH7.8 binding; however, their biochemical data cannot explain why IpaH7.8 ubiquitinates human but not mouse GSDMD as IpaH7.8 bound both human and mouse GSDMD, and displayed an even stronger binding affinity to mouse than human GSDMD (194 nM for human vs. 47 nM for mouse GSDMD) in their study. In our study, we found that IpaH7.8 actually does not bind mouse GSDMD. Structure of GSDMB in complex with IpaH7.8 identified key residues that are responsible for recognition of GSDMs by IpaH7.8. These residues are not structurally conserved in mouse GSDMD, thus explaining the species specificity of *Shigella* targeting human and non-human primates but not rodents. Moreover, we showed that IpaH7.8 is able to inhibit the activity of GSDMB independent of its E3 ligase activity. This is so far the first protein that is known that directly inhibits GSDM through binding. Lastly, we identified several lysine residues in the transmembrane region of GSDMB. Ubiquitination of these lysines by IpaH7.8 compromised the pore-forming activity of GSDMB. Thus, our finding identified another type of post-translational modification that regulates GSDM activity, in addition to the previously

reported itaconation of GSDMD and palmitoylation of GSDME, and so on. Therefore, we believe that our findings provide many new exciting insights into GSDM-mediated pyroptosis and bacterial pathogenesis.

As it pertains to the structures of full-length GSDMB and the GSDMB pore, aside from the specific structural roles played by alternative splicing, these structures confirm many of the regulatory mechanisms already reported for available structures of GSDMD and GSDMA3. For example, like GSDMD/A3, GSDMB overlaps significantly in the structural elements that determine its auto-inhibitory state and pore-assembly.

Response: We agree that the determined structures of GSDMA3 and GSDMD have provided many details of regulatory mechanisms of GSDM pore-formation. Notably, our structure of GSDMB pore presented here identified a critical lipid binding site (BS4) in the interdomain linker that is previously not considered to participate in the pore formation. Interestingly, BS4 is required for pore formation not only by GSDMB but also by GSDMD (Fig. 5). In GSDMB, BS4 is coded by exon 6, which is regulated by alternative splicing. Splicing generates at least 6 isoforms of GSDMB that exhibit distinct pore-forming activities. Only isoforms that contains structurally conserved BS4, such as isoforms 4 and 6, show normal pyroptotic activity. Our structure here highlights the functional role of exon 6/BS4 and the potential physiological importance of alternative splicing-mediated regulation of GSDMB.

An important unanswered question is whether cells exert control over the patterns of GSDMB isoform expression to achieve specific physiological functions. For example, in the pioneering report by Zhou et al., IFN- γ was found to induce GSDMB expression. In light of the isoform-specific activities of distinct GSDMB isoforms, it would be interesting to examine how their relative levels of expression change in response to IFN- γ stimulation. By extension, are there unidentified pathways that differentially regulate distinct GSDMB isoforms? Finally, do cancer cells differentially regulate GSDMB isoforms as a resistance mechanism against NK/T-cell tumor immunity?

Response: We have tested the expression of GSDMB isoforms in HeLa cells treated with IFN- γ . GSDMB isoforms 4 and 6 were detected in HeLa cells, however, IFN- γ treatment didn't have any effect on their expression (see below the Fig.1). This could be because IFN- γ -inducible expression of GSDMB may exists only in certain types of epithelial cells and cancer cell lines, such as A549, HT-29, SKBR3, and SW403 (Zhou, et al. 2020). It will be interesting to investigate the expression profile of GSDMB isoforms in these cell lines upon IFN- γ stimulation. However, we believe that this aspect and its relevance to cancer cell resistance to NK/T-cell tumor immunity is beyond the scope of the current study, whose focus is on the elucidation of structural determinants of GSDMB pore formation and its targeting by IpaH7.8. Nonetheless, we have included a brief discussion of this aspect in the text (see below):

“In addition to bacterial infection, GSDMB is associated with various cancers. A recent study indicated that expression of non-pyroptotic isoform 2 was higher than other isoforms in breast cancer patients³. Upregulation of isoform 2/3 may promote tumorigenesis and metastasis, leading to a poor overall survival of patients, while high expression of pyroptotic isoform 4 has an opposite effect³. Considering this potential association between GSDMB isoforms and cancer survival, additional studies are warranted to explore if cancer cells exploit differential expression of GSDMB isoforms to resist the attack by cytotoxic lymphocytes.”

2. How do GSDMB-N isoforms affect bacterial viability when added exogenously? i.e. expression from within bacteria is non-physiological

Response: We have tested the bacterial killing activities of GSDMB-N isoforms on *E.coli* DH5a by adding full length GSDMB and GZMA proteins exogenously to the bacterial culture. Upon cleavage by GZMA, GSDMB isoforms 4 and 6 showed strong bactericidal activity, while GSDMB isoform 1 is less toxic to bacteria. No significant bactericidal activity was observed for Isoforms 2 and 3. The results have been included in the revised Extended Fig. 7d. We also replaced Fig. 3c with a new figure (New Fig. 3c) in which GSDMB was added exogenously. GSDMB-mediated bacterial killing is significantly reduced in the presence of wild-type *Shigella* IpaH7.8, but not the two IpaH7.8 mutants, Y165E/Y166E and R186E/R228D.

3. Do Exon 6 coding sequences also mediate interactions between adjacent GSDMB monomers in the pore? If so, how do corresponding mutations impact GSDMB pore-forming activity?

Response: Yes, exon 6-coded sequence mediates interactions between adjacent GSDMB monomers in the pore. Exon 6 codes a canonical sequence ((NIHF)₁(RGKTK₂SFPE)₂) in GSDMB isoforms dividable into Region 1 and Region 2. The corresponding region 1 in GSDMD has been known to participate in subunit interaction in the pore (Extended Data Fig.10a, Liu, et al. 2019). In the structures of GSDMB and GSDMD pores, Region 1 contacts directly the helix α 3' from the adjacent subunit in the pore (Extended Data Fig.10a-c). Mutations of Region 1 in both GSDMB isoform 4/6 (“NIHF” to “GGGG”) and isoform 1 (“AGLD” to “GGGG”) almost completely abolished their liposome leakage activities (Extended Data Fig.10d), indicating the role of Region 1 in mediating GSDMB/D pore oligomerization. We have included these results in the revised manuscript, please see the new Extended Data Fig.10.

4. How does mutation of GSDMD basic patch 4 impact its pore-forming activity?

Response: Mutation of basic patch 4 in GSDMD also significantly reduced the pore-forming activity. We have tested the mutation both *in vitro* and in cell. Please see new Fig. 5g and h and Extended Data Fig. 10f and g.

5. Does the purified mouse GSDMD undergo cleavage by CASP4/11? If so, does this unleash its pore-forming activity?

Response: Yes, purified mouse GSDMD can be cleaved by active caspase-11 *in vitro*. Cleavage caused significant leakage of liposomes containing acidic phosphatidylserine (PS) (see below Fig.2, and our and other's previous studies, Liu, *et al.* 2016; Kayagaki, *et al.* 2015; Ding, *et al.* 2016, etc). Caspase-4/11 specifically recognizes and cleaves the interdomain linker ("²⁷²LLSDG²⁷⁶") in mouse GSDMD. Cleavage liberates the N-terminal domain of mouse GSDMD, which specifically binds the acidic lipids in the inner leaflet of the plasma membrane to form pores that cause pyroptotic cell death.

We thank Reviewer #2 for the insightful comments and suggestions.

Referee #2 (Remarks to the Author):

In this manuscript, Wang et al. reported the cryo-EM structures of both GSDMB N-terminal domain (NT) pore, and the complex of full-length GSDMB with the Shigella virulence effector IpaH7.8. The structure of the GSDMB-IpaH7.8 complex revealed that a three-negatively-charged-residue motif determines the recognition of GSDMB-NT by IpaH7.8, which explains the recognition of GSDMB and human GSDMD, and the lack of interaction with mouse GSDMD. Interestingly, the authors showed that the binding of IpaH7.8, as well as the ubiquitination of GSDMB (in residues within the transmembrane domain), directly inhibits GSDMB pore formation. This is by contrast to GSDMD whose ubiquitination by IpaH7.8 was shown previously not to be sufficient for inhibition of pore formation, but whose inhibition requires subsequent proteasomal degradation. The authors investigated the different GSDMB isoforms and showed that they exhibited different pore formation ability. The GSDMB pore structure they solved further revealed that the interdomain linker, which varies between isoforms, participates in lipid binding, which can explain the isoform-dependent differential activities. These studies are carefully performed and of high quality.

Specific comments:

Major:

1- Several in-vitro ubiquitination experiments such as Figure 2C, 2E, 2F, 2G, 2H, 2I and 2J should show the bottom part of the gel to demonstrate the unconjugated ubiquitin in the reaction. This is important for GSDMB mutations with decreased ubiquitination such as Y165A/Y166A or Y165E/Y166E, lanes 4 and 5 in Figure 2E. This information will prove that the ubiquitin was equally available in all reactions but was not conjugated in the case of the mutants. Ubiquitin is an 8.6 kDa protein and should be visible in a gradient gel SDS-PAGE.

Response: We thank the reviewer for the comments and suggestions. We have re-run the samples in these figures with 8-15% gradient gels. The new gels clearly showed that ubiquitins were excess in all the reactions and were equally available. The decrease in ubiquitin levels was also observed in those reactions where ubiquitination occurred.

2- Related to Figure 2F: The authors used a co-transfection system and observed loss of IpaH7.8-mediated degradation of certain GSDMB mutants. This observation should be supplemented with ubiquitin blot showing ubiquitinated species on immunoprecipitated GSDMB as a second line of evidence that the proposed mutations alter IpaH7.8-mediated degradation of GSDMB.

Response: We thank the reviewer for this suggestion and we examined ubiquitination of GSDMB by IpaH7.8 as suggested. Briefly, HEK293T cells were co-transfected with plasmids of FLAG-GSDMB and empty vector or HA-IpaH7.8 (wild-type or indicated mutants). Bortezomib was used to prevent proteasomal degradation of ubiquitinated proteins. IP samples immunoprecipitated using anti-FLAG beads were immunoblotted with anti-ubiquitin and anti-FLAG antibodies. Immunoblots showed ubiquitination of GSDMB by wild-type IpaH7.8, and importantly, significant reduction in ubiquitination level was observed when GSDMB was co-expressed with IpaH7.8 mutants. We have presented this data in the new Fig. 2f, and moved the previous Fig. 2f to the Extended Data Fig. 3c.

3- Figure 2J lacks negative control for the poly-ubiquitination signal. To show specificity, catalytically inactive form of IpaH7.8 should be included in the analysis.

Response: We have repeated the experiment with the IpaH7.8^{C357A} inactive mutant as a control as suggested (Fig. 2j).

4- 3D-classification of cryo-EM datasets includes also the prepore structure of GSDMB. Models of the prepore structure should also be built and reported.

Response: We thank the reviewer for pointing this out. Unfortunately, the resolution of the GSDMB prepore is very low (around ~9.6 Å). The low resolution is probably because of the less particle numbers and heterogeneity of the prepore (lower resolution of prepores in both GSDMA3 and human GSDMD, Ruan, et al, 2018; Xia, et al, 2021). We have built the model by rigidly fitting the auto-inhibited GSDMB-N in the cryo-EM density. Enlarged cryo-EM density and model are presented in the revised Extended Data Fig. 9. However, we prefer not to deposit/report and discuss this structure because of the low accuracy of the model.

Minor

1- It should be mentioned in the legend of Figure 3G that the demonstrated image is a Coomassie blue-stained SDS-PAGE as it was mentioned in other figure legends.

Response: We thank the reviewer for pointing it out. We have added this information to the figure legend.

2- The sequence arrangements in mouse GSDMD used in Figure 2H should be illustrated with a representative schema.

Response: We have added an upper panel in the Fig. 2h showing the sequence alignment to make it easier to follow.

3- 3D-classification of GSDMB was referred in the text as Extended Figure Data 7C, however, this data is in the next figure (Extended Figure Data 8C). This should be corrected in the text.

Response: We thank the reviewer for pointing it out. We have corrected these errors.

We thank Reviewer #3 for the insightful comments and suggestions.

Referee #3 (Remarks to the Author):

This paper discusses how a bacterial secretion system effector protein, Ipa7.8 from *Shigella*, binds to the human GSDMB protein. GSDMB is a member of the GSDM family of proteins, of which the most famous is GSDMD. These are proteins that are processed, and one cleavage fragment oligomerizes in the cell membrane to create a pore. GSDM pores are consistent with pyroptotic cell death. The biology of GSDMB is not clear, but it is believed that Granzyme B can cleave it for pore formation. *Shigella* Ipa7.8 is in the present report shown to interact with GSDMB in cryo-EM experiments. They also investigate the role of Ubiquitination of GSDMB. In addition, structures of the whole GSDMB pore is clarified, together with studies of various splicing variants of GSDMB. The effect of these aspects on liposome pore formation is shown. Taken together, these defined data sets represent a significant advance of the field.

Specific comments:

Overall, the data are clear and it is very interesting to observe the direct binding of a bacterial inhibitor to a GSDM protein, this is a first. It is also a strength to have meaningful discussions about various GSDMB isoforms, this aspect of splicing is often neglected in many papers, although this paper could have covered more ground on various isoforms. How are these isoforms compare to what has been studied in previous papers?

Response: Though there are many published studies on GSDMB, very few of them have mentioned the isoform(s) that was used in their study. By surveying the length of the protein/gene or the gene ID if mentioned in the publication, we summarize that:

1. Ding, *et al.* (2016) and Zhou, *et al.* (2020) used isoform 4 (484aa, the longest isoform) in their study. Isoform 4 showed normal pyroptotic activity upon cleavage by GZMA.
2. Shi, *et al.* (2015) and Chen, *et al.* (2019) showed that GSDMB exhibited no pyroptotic activity, but the specific GSDMB isoforms that were used in these studies were not mentioned.
3. Hansen, *et al.* (2022) and Rana, *et al.* (2022) used isoform 1 in their study. Isoform 1 was cytotoxic to bacteria, but did not induce pyroptosis.
4. A recent study by Oltra, *et al.* studied many constructs of GSDMB, and found that the construct 1-220 (equivalent to isoform 2) showed no pyroptotic activity and construct 1-242 (equivalent to isoform 4) and construct 1-233 (equivalent to isoform 6) showed comparable pyroptotic activity to GSDMD. The isoform 1 with an “AGLD” insertion in the interdomain linker was not examined in this study.
5. There are other publications that studied different isoforms without focusing the pore-forming activities, including the studies of Sun and Yang, *et al.* (2008), Hergueta-Redondo, *et al.* (2014), Li, *et al.* (2021), *etc.*

In our study, we systematically studied all 5 isoforms (1, 2, 3, 4, and 6) of GSDMB containing N terminal domains (isoform 5 contains only C-terminal domain). Our results are in overall consistent with the previous studies. Isoforms 4 and 6 showed strong pyroptotic activity and isoform 1 showed attenuated pore-forming activity *in vitro* but was unable to induce pyroptotic cell death in cells, while isoforms 2 and 3 showed no activity both *in vitro* and in cells.

Many of their observations are backed up with experiments with lysis of liposomes. However, I do miss a more direct assay for actual cell death. They express GSDMB in 293 cells for WB/IP type

of experiments, but why are 293 cells not also used for cell death assays? In the right conditions, over-expression in 293 is a relatively quick and easy way to induce cell death. This type of assay, although still in an artificial system, would carry more weight and could lend further support to many of their conclusions on Ipa7.8 inhibition, isoforms etc, and this could be important.

Response: We have added these cellular experiments in the revised manuscript.

1. We have tested the cytotoxicity of GSDMB isoforms in HEK293 cells. Cell death was monitored by Hoechst/PI double staining. The results showed that isoforms 4 and 6 induced remarkable cell death, while isoforms 1, 2, and 3 did not. These data are included in the new Fig 4b and Extended data Fig. 7d.
2. We also tested the IpaH7.8 inhibition of GSDMB in cells. We co-transfected HEK293T cells with GSDMB isoform 4, which is capable of inducing pyroptosis, and IpaH7.8 (WT or C357A mutant). The results showed that GSDMB isoform 4 mediated pyroptotic cell death was greatly inhibited by IpaH7.8. The inhibition is IpaH7.8 E3 ligase activity independent. These data confirm the direct inhibition of GSDMB by IpaH7.8 through binding. These data are included in the new Extended data Fig. 7e and f.
3. We also tested the effect of mutation of lipid binding site 4 (BS4) in HEK293T cells. Consistent with the *in vitro* liposome leakage assay, the mutation of BS4 significantly reduced the capability of both GSDMB and GSDMD to induce cell death. These data are included in the new Fig 5h and Extended data Fig. 10f and g.

Methodology seems valid. Appropriate statistical analyses are included. References are OK. Conclusions are well supported.

Response: We thank the reviewer for the positive comments.

Fig 2 c: “single mutations of either E15 or D17 in GSDMB into a reversely 132 charged or uncharged residue completely or significantly abolished the ubiquitination of GSDMB 133 by IpaH7.8 *in vitro*”

D17S seems more of a partial reduction, especially since Gsdmb band appears significantly reduced like WT. Perhaps the authors can be more careful with the language here.

Response: We thank the reviewer for pointing this out. We have rephrased this sentence: “single mutations of either E15 or D17 in GSDMB into a reversely charged or uncharged residue significantly or partially reduced the ubiquitination of GSDMB by IpaH7.8 *in vitro*”.

Fig 3c : The bacterial growth inhibition assay was nice as another measure of GSDMB pore function. However it is quite artificial to place GSDMB into E coli. Exploring Gdsmb from the host (or artificial host) inserting into bacterial membranes would be great support.

Response: We have tested the bacterial killing activities of GSDMB-N isoforms on *E.coli* DH5a by adding full length GSDMB and GZMA proteins exogenously to the bacterial culture. Upon cleavage by GZMA, GSDMB isoforms 4 and 6 showed strong bactericidal activity, while GSDMB isoform 1 is less toxic to bacteria. No significant bactericidal activity was observed for Isoforms 2 and 3. The results have been included in the revised Extended Fig. 7d.

We also replaced Fig. 3c with a new figure (New Fig. 3c) in which GSDMB was added exogenously. GSDMB-mediated bacterial killing is significantly reduced in the presence of wild-type *Shigella* IpaH7.8, but not the two IpaH7.8 mutants, Y165E/Y166E and R186E/R228D.

The Ub-studies, in particular how they relate to species differences in GSDMB proteins, is a nice addition, especially the “rescue” of mouse GSDMB with human sequences.

Response: We thank the reviewer for the positive comment.

Pore analysis: What is the reasoning behind the choice of isoform 1 for the Cryo-em pore analysis instead of 4 or 6 that had better permeabilization activity? Do they think other isoforms could give different pore compositions? Please address this question. “We primarily focused on GSDMB isoform 1 because of its attenuated pore-forming activity”. But why is that better comparison to other GSDM proteins?

Response: We thank the reviewer for the comment. To make our choice of GSDMB isoform 1 more reasonable, we revised the manuscript by adding negative staining images of pores formed by isoforms 4 and 6 and compared them with pores of isoform 1. The pores formed by isoform 1 showed better behaviour with less aggregation as compared to those of isoforms 4 and 6 (Extended Fig. 8 and b). The text has been modified as follows.

“To address why GSDMB isoforms exhibit distinct pore-forming activities, we sought to determine the cryo-EM structure of the GSDMB pore. GSDMB pore was reconstituted by cleaving the full-length GSDMB on CL-liposomes^{9,21}. GSDMB pore-loaded liposomes were then solubilized in detergent C12E8 followed by size-exclusion chromatography purification. Screening of GSDMB isoforms identified that isoforms 1, 4, and 6 formed pores similar in size and shape. However, the pores of isoform 1 showed less aggregation than those of isoforms 4 and 6 and isoform 1 pores distributed evenly on cryo-EM grids (Extended Data Fig. 7a, b). We thus subjected GSDMB isoform 1 to the cryo-EM analysis.”

Please include a discussion on the potential importance of GSDMB isoforms on *Shigella* infection.

Response: We have added a paragraph discussing the potential importance of GSDMB isoforms in bacterial infections. See below:

“The cryo-EM structure of the GSDMB pore illustrates the pore-forming mechanism of GSDMB and reveals a unique regulatory mechanism in controlling pore-forming activities of GSDMB isoforms through the interdomain linker, which participates in both pore oligomerization and lipid binding. Alternative splicing of the GSDMB transcript generates at least six isoforms varying in their interdomain linkers²⁴. As the interdomain linker is vital for the pore formation, GSDMB isoforms exhibit distinct pore-forming activities. GSDMB is widely expressed in various cell types and tissues^{5,38,39}, where its isoforms may be differentially regulated. Such differences in the expression and activities of GSDMB isoforms probably represent a fight-back mechanism in host against *Shigella* infection which is benefited from the IpaH7.8-mediated inhibition of GSDMB. We hypothesize that pyroptotic isoforms 4 and 6 might be dominant in epithelial cells and the activation of isoform 4/6 during bacterial infection triggers the pyroptotic cell death of infected epithelial cells, thus eliminating the replicative niche of intracellular bacteria pathogens, such as *Shigella*. While non-pyroptotic isoform 1 might be dominant in macrophages or dendritic cells where it targets and kills cytosolic *Shigella*—instead of inducing pyroptosis—thus ensuring the survival of these antigen presenting cells for T cell activation. Nonlytic isoforms 2 and 3 might be expressed in both epithelial cells and immune cells, ensuring the functionalities of isoforms 1, 4, and 6 by competitively binding to the *Shigella* IpaH7.8. Currently, cell-specific distribution,

abundance, and function of each GSDMB isoform are not well understood. Physiological relevance of GSDMB isoforms to antibacterial immunity need to be further investigated.”

Reviewer Reports on the First Revision:

Referees' comments:

Referee #1 (Remarks to the Author):

The authors suggest that GSDMB plays a critical role in innate immunity to Shigella ("... activities of GSDMB isoforms probably represent a fight-back mechanism in host against Shigella infection.. ") The claim of the importance of GSDMB in immunity to Shigella infections is undermined by the finding that the gene encoding GSDMB, a supposed protector against Shigella, is present in humans who are sensitive to less than ten Shigella organisms, but absent in mice who are resistant to over a million Shigella! Surely, the opposite relationship would have been expected if GSDMB played any role in immunity to Shigella. This should be discussed in the very least.

Referee #2 (Remarks to the Author):

This reviewer is satisfied with the revision.

Referee #3 (Remarks to the Author):

Thanks for addressing reviewer comments.

Author Rebuttals to First Revision:

Referee #1 (Remarks to the Author):

The authors suggest that GSDMB plays a critical role in innate immunity to *Shigella* ("... activities of GSDMB isoforms probably represent a fight-back mechanism in host against *Shigella* infection.. ") The claim of the importance of GSDMB in immunity to *Shigella* infections is undermined by the finding that the gene encoding GSDMB, a supposed protector against *Shigella*, is present in humans who are sensitive to less than ten *Shigella* organisms, but absent in mice who are resistant to over a million *Shigella*! Surely, the opposite relationship would have been expected if GSDMB played any role in immunity to *Shigella*. This should be discussed in the very least.

Response: We thank the reviewer for pointing this out. We meant to state GSDMB and GSDMD are generally important for innate immunity to bacterial pathogens. However, a pathogen like *Shigella* interferes with human GSDMB and GSDMD to minimize their contribution to host defense, which may explain human susceptibility to *Shigella* infection. On the other hand, murine GSDMD is not sensitive to *Shigella* IpaH7.8 and it can still mediate host resistance to *Shigella* in mice. However it is worth noting that people with shigellosis usually recover in 5 to 7 days without needing antibiotics³⁴. Though targeted by IpaH7.8, GSDMB and GSDMD may still play a role in rendering *Shigella* infection in humans self-limiting. We have added this discussion in the revised manuscript.

Referee #2 (Remarks to the Author):

This reviewer is satisfied with the revision.

Referee #3 (Remarks to the Author):

Thanks for addressing reviewer comments.